# Stochastic modelling of delays and buffering in 5G-IoT ecosystems with programmable P4 switches based on BMAP

Viacheslav Kovtun[1]*, Maria Yukhimchuk[1], Jamil Abedalrahim Jamil Alsayaydeh[2]*, Dinara Berdysheva[3]

1 Department of Computer Control Systems, Faculty of Intelligent Information Technologies and Automation, Vinnytsia National Technical University, Vinnytsia, Ukraine, 2 Department of Engineering Technology, Fakulti Teknologi and Kejuruteraan Elektronik and Komputer (FTKEK), Universiti Teknikal Malaysia Melaka, Durian Tunggal (UTeM), Melaka, Malaysia, 3 Department of Computer Science, Al-Farabi Kazakh National university, Almaty, Kazakhstan

* kovtun_v_v@vntu.edu.ua (VK); jamil@utem.edu.my (JAJA)

## Abstract

This study presents a hybrid stochastic model for evaluating delays and buffering in 5G-IoT ecosystems with programmable P4 switches, where traffic patterns exhibit strong batch-like properties. The proposed approach integrates a batch Markovian arrival process (BMAP) with a phase-type service structure and semi-Markov modelling of control-plane interactions, thereby capturing both the temporal variability of IoT traffic and the hybrid nature of routing logic. Analytical expressions for the expected processing time and queue length were derived using extended G/G/1, $H_2/H_2/1$, M/G/1, and M/N/1 queueing frameworks. Unlike traditional queueing models, the proposed framework is the first to simultaneously incorporate BMAP-driven bursty arrivals, phase-type service distributions, and semi-Markov representation of control-plane interaction dynamics. This integrated design enables more accurate characterisation of real IoT traffic and significantly improves predictive accuracy. The model was validated on real-world traffic datasets, demonstrating that BMAP more accurately reflects the structure of IoT traffic than classical Poisson or MMPP models. Notably, the BMAP-based approach reduced the modelling error by up to 38% compared to Poisson-based approximations and by 22% compared to MMPP-based ones under bursty traffic conditions. Simulation results confirm that increasing the control-plane involvement probability from 0.2 to 0.7, under a fixed average batch size of 12 requests, leads to a 2.6-fold increase in processing delay. Furthermore, the $H_2/H_2/1$ model showed the highest alignment with empirical data, accurately reflecting the multi-phase service structure and control flow saturation effects. Additional 3D analyses revealed strong nonlinear dependencies of delay on the batchiness factor, dispersion in processing times, and phase asymmetry parameters.

**Data availability statement:** The data underlying the results presented in the study are available from the Kaggle repository: IoT Traffic Generation Patterns Dataset [https://www.kaggle.com/datasets/tubitak1001118e277/iot-traffic-generation-patterns].

**Funding:** The author(s) received no specific funding for this work.

**Competing interests:** The authors have declared that no competing interests exist.

# 1. Introduction

## 1.1. Relevance of the research

The rapid development of wireless communication technologies and the large-scale deployment of intelligent devices have driven the exponential growth of Internet of Things (IoT) ecosystems [1]. One of the key drivers of this transformation is the implementation of 5G networks, which significantly enhance the scalability, responsiveness, and adaptability of IoT infrastructures [2, 3]. According to the IoT Analytics, the number of connected IoT devices reached 16.6 billion in 2023, representing a 15% increase compared to the previous year. It is projected that this figure will reach 18.8 billion by the end of 2024 and will exceed 40 billion by 2030, as a result of ongoing digitalisation across industries such as manufacturing, healthcare, logistics, agriculture, and urban infrastructure.

This dynamic increase in the number of sensor, actuator, and edge devices introduces new challenges for network infrastructure, particularly in terms of latency, throughput, and the flexibility of traffic processing. The wide range of modern applications (from autonomous transport and industrial automation to telemedicine and augmented reality systems) demands ultra-low data transmission latency with guaranteed Quality of Service (QoS) [4,5]. While 4G LTE networks typically provide latencies of 30–50 ms, 5G technologies can reduce this figure to below 10 ms, and, with the activation of Ultra-Reliable Low-Latency Communication (URLLC), even to 1 ms under controlled conditions [6]. In addition to low latency, a critical requirement is the network's adaptability to unpredictable traffic changes. Typical IoT traffic is highly heterogeneous, periodic or event-triggered, resulting in bursts of messages with a clustered structure. This uneven arrival pattern complicates the use of classical queuing and scheduling methods and instead necessitates dynamic reconfiguration and context-aware routing [7]. Moreover, the 5G architecture supports massive Machine-Type Communication (mMTC), capable of serving up to one million devices per square kilometre [8,9], opening new opportunities for large-scale deployments in the segment of low-power, low-data-rate devices. However, this very scalability contributes to increased variability in both the rate of arrivals and the intensity of request processing, particularly in programmable networks involving P4 switches and analytical modules. Thus, in the context of the convergence of 5G and IoT technologies, the modelling and control of delays, as well as the adaptive processing of traffic, are no longer merely matters of optimisation, but rather essential prerequisites for ensuring the stability, flexibility, and scalability of network services. This necessitates the development of new analytical and stochastic models capable of accurately capturing the complex behaviour of traffic and the dynamics of network systems. Such complexity arises not only from the volume of transmitted data, but also from the specific structural features of IoT traffic, its bursty nature, temporal clustering, and the interplay between local and centralised decision-making. These factors lead to phased processing workflows and hybrid routing, which challenge conventional modelling assumptions. As will be further detailed in Section 1.2, classical queueing models offer limited analytical capability under such conditions, thereby motivating the integration of programmable and adaptive mechanisms into network infrastructure.

Contemporary challenges associated with the processing of unpredictable, clustered traffic in 5G-IoT ecosystems have necessitated a shift from rigidly predefined routing logic to programmable network solutions. In this context, programmable P4 switches play a particularly important role [9,10]. These network devices enable adaptive traffic management at the data link and network layers, in accordance with current conditions, service policies, and dynamic analytics. The P4 (Programming Protocol-independent Packet Processors) programming language [11] enables users to define packet processing logic in the form of match-action tables, which are implemented directly in the switch's hardware. This architecture provides protocol independence, dynamic reconfigurability of switching logic, and support for complex routing scenarios in real time. In 5G-IoT environments, in particular, it opens up opportunities for data burst aggregation, congestion detection, adaptive queue management, and prioritisation based on quality of service (QoS) requirements. In contrast to traditional switches with fixed functionality, P4 switches can be reprogrammed dynamically in response to changes in topology, traffic intensity, or service requirements. When integrated with analytical modules or Software-Defined Networking (SDN) controllers, these devices enable distributed or hybrid decision-making logic, wherein part of the processing is executed locally and part centrally. This configuration achieves a balance between responsiveness and adaptability, which is critically important for IoT applications characterised by high traffic variability. So, programmable P4 switches serve as a key technological component within 5G-IoT infrastructure, enabling the development of intelligent routing policies that adapt to real transmission conditions, support QoS-oriented traffic processing, and contribute to the overall enhancement of network ecosystem performance [9,10]. At the same time, the increased flexibility and programmability of such network components introduce additional layers of complexity in the evaluation and management of system performance. The ability to adapt routing logic dynamically, combined with the heterogeneous and bursty nature of IoT traffic, necessitates a refined approach to the modelling of delays and buffering. Rather than relying on static or oversimplified assumptions, modern 5G-IoT environments require stochastic models capable of representing clustered message arrivals, phased processing procedures, and hybrid interaction between decentralised switches and centralised controllers.

Traffic in 5G-IoT ecosystems is generated not only by continuously active streams, but primarily through periodic, event-triggered, or synchronised data transmissions. This results in the formation of message bursts arriving at network nodes in clustered patterns, interspersed with periods of inactivity. Such behaviour is typical, for example, in sensor networks, monitoring systems, smart meters, or surveillance cameras. Special attention must be given to the interaction between P4 switches and analytical modules or controllers, which define the processing rules for new flows. A portion of requests, for which no predefined routing rules exist, requires additional analysis at the control plane level, leading to the formation of a parallel queue of service messages. Delays in this queue contribute to the overall processing time of requests, particularly under conditions of network congestion. Thus, realistic modelling of delays and buffering in 5G-IoT environments requires consideration of multiple factors: the bursty nature of traffic, the phase structure of processing, the interaction between the data plane and control plane, as well as stochastic fluctuations in arrival and service intensities. This creates an objective need for the development of advanced analytical models capable of adequately representing the multifactor dynamics of IoT traffic and the associated behaviour of queues and delays, as further discussed in Section 1.2.

## 1.2. State-of-the-art

Against the backdrop of the rapid increase in variable IoT traffic, there emerges a pressing need for adequate modelling of delays arising in software-defined network (SDN) environments, particularly when programmable P4-based switches are employed. In the classical SDN architecture [12, 13], traffic processing follows two scenarios: local handling of packets at the data plane (when an appropriate flow-entry exists in the switch's table), or redirection of the request to the controller (control plane) to obtain processing instructions. This approach creates significant dependencies between delay and the flow table hit probability, as well as the overall intensity of requests directed to the controller.

The study in [14] presents an analytical model of SDN networks with Priority Queues (PQ), which accounts for the bursty nature of traffic, modelled using a Poisson process modulated by a Markov chain (MMPP). This model enables

the evaluation of average delay, taking into consideration the flow table hit probability, the limited buffer sizes at both the switch and the controller, as well as varying service rates. It was demonstrated that even with a 50% hit rate ($\varepsilon = 0.5$), delay is significantly influenced by the distribution of computational resources between the SDN switch and the controller: prioritising the switch ($\mu_s > \mu_c$) considerably reduces average delay, whereas overloading the controller creates a bottleneck in request processing.

A distinctive feature of the PQ model is the presence of two queues – a low-priority queue for regular traffic and a high-priority queue for control messages from the controller. This structure allows for an accurate representation of the system's real behaviour, in which requests lacking predefined flow entries experience a two-phase delay: first, waiting for transmission to the controller, and then awaiting the return of processing instructions and subsequent rehandling of the packet.

Studies [15] indicate that classical SDN architectures suffer from critical delays in scenarios involving the mass arrival of new flows, particularly in distributed IoT deployments. In a network comprising 100 switches, the volume of control traffic may reach 10 million requests per second, placing an unsustainable load on controllers – even those with multithreading support. Under such conditions, the only viable solution is a transition to a decentralised control-plane architecture (e.g., ONOS or Kandoo), which enables local controllers to manage sub-networks, thereby minimising latency and limiting the scope of signalling.

Moreover, models incorporating MMPP arrival processes and PQ servicing open new possibilities for accurately modelling average delay, queueing time within individual queues (data plane/ control plane), buffer blocking probabilities, and overall system throughput [14]. For example, increasing the switch's service rate from 5 to 40 packets per second reduces the average system delay from 16 to 3 seconds, demonstrating the effectiveness of shifting computational load to the data plane.

In the traditional approach to modelling delays in SDN, M/M/1-type models with exponential inter-arrival and service times are most commonly used [14, 16, 17]. Despite their mathematical simplicity, such models fail to capture the fundamental characteristics of IoT traffic, particularly its clustered nature, variability, and phased processing. As demonstrated in [17], even in basic SDN architectures, applying M/M/1 models to both the data plane and control plane results in underestimated delay values and does not reflect the alternation between control and user traffic flows. In more complex scenarios, such as P4 environments with combined (hybrid) routing, these models become entirely unsuitable.

The M/G/1 model [18, 19], which allows for an arbitrary service time distribution, can partially reflect differences in processing depending on QoS class, the complexity of P4 programs, or the load on the switch. However, the assumption of Poisson arrivals renders the model sensitive to inaccuracies in cases involving burst transmissions. As highlighted in [19], M/G/1 is unable to capture scenarios involving batch queue formation, which typically occur during mass sensor activations or at peak load points in 5G-IoT environments.

More flexible are MMPP/M/1 models, which allow for the modelling of time-varying packet arrival intensities – such as the transition of devices from inactive to active states [20]. In the context of SDN with priority queues, such models enable the representation of delay as a function of the flow table hit probability. However, constructing even a simple two-phase system (e.g., switch + controller) requires a complex multi-component decomposition. Researchers in [21] were compelled to apply the Empty Buffer Approximation method, which yields only approximate results and performs particularly poorly under high-load conditions – for instance, with hit probability $\varepsilon \to 0$, the average delay exceeds 15 seconds at an arrival rate of 50 packets per second.

The Batch Markovian Arrival Process (BMAP), unlike the previously discussed models, enables the representation of batch traffic arrivals, which are typical of periodically synchronised sensors. Its potential for SDN applications is considerable, particularly in scenarios where processing depends on the class or criticality of the request. However, as demonstrated in [22], even in complex SDN-IoT environments, BMAP is most often employed in isolation and without

consideration of multi-stage processing or hybrid routing, which limits its applicability for comprehensive delay modelling. Moreover, BMAP-based models frequently remain at the simulation level, lacking analytical generalisation, thereby preventing their direct use in QoS optimisation.

Phase-type models, particularly $H_2/H_2/1$, are potentially the most suitable for describing the real processing workflow in SDN with P4 switches, as they allow for sequential handling in the data plane, redirection to the control plane, and subsequent retransmission. However, as noted in the review [23], such models are rarely applied to SDN due to their highly complex parameterisation requirements (e.g., phase-specific service time distributions dependent on load), and they seldom yield closed-form analytical expressions for processing time or buffer loss probability.

## 1.3. Main attributes of the research

The object of the study is the process of stochastic modelling of delays and buffering in 5G-IoT ecosystems with programmable P4 switches under conditions of clustered traffic.

The subject of the study is stochastic models for the analysis of queues and delays in hybrid routing scenarios, taking into account QoS priorities, the bursty nature of arrivals, and the interaction between data-plane and control-plane components.

The aim of the study is to develop a generalised stochastic model that enables an adequate assessment of the average request processing time and queue length in 5G-IoT infrastructures with programmable network devices operating under hybrid routing logic.

To achieve the stated aim, the following objectives are set:

1. To analyse the architecture of information interaction in 5G-IoT systems with P4 technology support;

2. To formalise the processes of request arrival and processing, taking into account the bursty nature of traffic;

3. To construct an analytical model of stochastic interaction between P4 switches and the analytical control module;

4. To evaluate the average request processing time and queue length under hybrid routing conditions, using G/G/1 and $H_2/H_2/1$ models.

The main contribution of this study is the development of a hybrid analytical and simulation-based model that accurately characterises delays and buffering processes in 5G-IoT ecosystems featuring programmable P4 switches. This model uniquely integrates a batch Markovian arrival process (BMAP), phase-type service distributions, and a semi-Markov representation of control-plane interactions. By extending classical queueing systems (G/G/1, $H_2/H_2/1$, M/G/1, M/N/1) with stochastic feedback between the data and control planes, the research provides novel analytical expressions for mean processing time and queue length under realistic traffic conditions. The model's validity is confirmed through empirical data, with the $H_2/H_2/1$ configuration and BMAP arrival patterns demonstrating superior alignment with observed IoT behaviour compared to traditional Poisson-based approximations. These results offer a robust framework for optimising QoS-aware hybrid routing and resource allocation in programmable, delay-sensitive network infrastructures.

The adequacy and practical value of the model are confirmed through numerical simulation and validation based on empirical characteristics of real IoT traffic from an open dataset. This substantiates the advantages of the proposed approach over classical models under bursty and highly variable load conditions. The results obtained can be used to optimise QoS parameters and enhance the efficiency of network service delivery in next-generation intelligent ecosystems with programmable components.

The subsequent sections of the article follow a consistent research logic encompassing the formalisation of the problem, the construction of an analytical model, its empirical validation, and the interpretation of results in the context of delay and buffering within 5G-IoT environments equipped with programmable P4 switches.

Section 2, "Models and Methods", presents a formalised description of the information exchange architecture between the P4 switch and the analytical module, defines the parameters of clustered arrivals and introduces a hybrid stochastic routing model that combines a BMAP input, phase-type $H_2/H_2/1$ distributions, and a semi-Markovian representation of interaction between the data and control planes. Subsection 2.2 focuses on the processing of requests within the analytical module using a semi-Markov approach, which involves queueing analysis under random service durations modelled by a truncated normal distribution. Subsections 2.3–2.4 generalise the model to scenarios involving hybrid routing and phase variability, incorporating the G/G/1 framework and its refinement via the $H_2/H_2/1$ scheme, which enables an adequate description of delays under multiphase request processing.

Section 3, "Results and Discussion", presents the empirical verification of the model using the open-access dataset IoT Traffic Generation Patterns (Kaggle): time series were constructed, histograms of clustering and inter-arrival intervals were generated, the density of inter-packet intervals was estimated using KDE, and the accuracy of service time distribution approximation was compared across $M/N/1$, $H_2/N/1$, and $H_2/H_2/1$ models (Fig 5). A series of three-dimensional visualisations (Figs 6–9) illustrates the dependence of average delay on service parameters, phase asymmetry, and processing time dispersion. Particular attention is given to an additional experiment conducted on traffic from four devices, which confirms the generalisability of the model beyond the core scenario (Figs 10–11). The analytical section concludes with Table 1, which presents a comparative analysis of the proposed approach against the most relevant state-of-the-art counterparts reported in the literature, thereby enabling an objective evaluation of its advantages in terms of modelling accuracy.

Finally, Section 4, "Conclusions", summarises the scientific findings, confirms the benefits of the combined model in capturing phase-sensitive and clustered traffic, and outlines prospects for its further application in QoS-oriented management within next-generation intelligent networks.

## 2. Models and methods

### 2.1. Statement of the research

A typical scenario of information interaction in a 5G-IoT ecosystem employing P4 technology is presented in the structural diagram (Fig 1). The upper section features the analytical module, which performs functions such as traffic monitoring, analysis of buffering and delays, congestion detection, and dynamic adjustment of service parameters. The lower section of the diagram depicts the 5G-IoT P4 switch, which implements programmable traffic processing logic, adapting forwarding rules according to instructions received from the analytical module. Interaction between the modules occurs via a bidirectional data exchange channel: upward communication conveys queue statistics and delay characteristics, while downward communication delivers control messages with policy parameters, processing order, and priority levels. The mathematical framework describing this information interaction (within the basis of queueing theory and accounting for the specific traffic structure) will be formalised in the following sections.

In modern 5G-IoT ecosystems, a significant portion of traffic is generated by end devices (sensors, meters, video cameras) exhibiting periodic activity or reacting to external events. Under such conditions, data is typically transmitted not as individual independent packets, but as groups (or bursts) formed in advance or triggered simultaneously – often as a

**Table 1. Delay and queueing characteristics under varying input conditions.**

| Mean Batch Size | Mean Service Time (ms) | Mean Delay E[T] (ms) | 95% CI E[T] (ms) | Mean Queue Length E[Q] | 95% CI E[Q] |
|---|---|---|---|---|---|
| 3 | 1.0 | 4.75 | [4.62, 4.89] | 4.69 | [4.53, 4.85] |
| 5 | 1.0 | 7.91 | [7.74, 8.08] | 7.85 | [7.64, 8.07] |
| 7 | 1.0 | 11.42 | [11.16, 11.69] | 11.36 | [11.04, 11.68] |
| 9 | 1.0 | 15.09 | [14.75, 15.43] | 15.02 | [14.60, 15.44] |
| 5 | 2.0 | 15.70 | [15.31, 16.09] | 15.60 | [15.15, 16.05] |

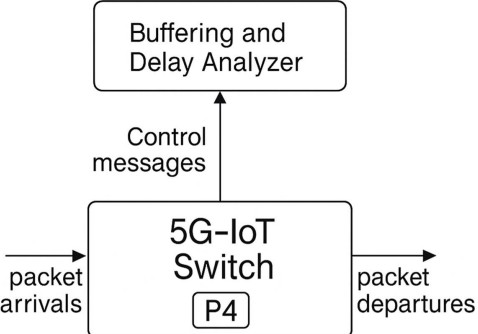

**Fig 1. Diagram of the information interaction scenario in a 5G-IoT ecosystem with integrated P4 technology.**

result of event-based activation. This is characteristic of buffered transmission scenarios, in which data is accumulated over a defined period (e.g., every five minutes) and then transmitted collectively. Contemporary P4-enabled switches are capable of aggregating or classifying incoming traffic according to predefined policies, further reinforcing the formation of batch arrivals. Consequently, in the context of a 5G-IoT ecosystem, the application of the BMAP arrival model allows for a more accurate representation of real traffic characteristics at the network level and ensures adequate queue modelling that accounts for correlations, intensity, and the bursty nature of events.

We consider the request processing procedure at a node within a 5G-IoT ecosystem equipped with P4 support. Each incoming packet (request) arriving at a programmable network device (e.g., a gNB or edge switch) undergoes preliminary analysis to identify its flow and to match an appropriate processing or forwarding rule within the existing match-action tables implemented via P4. If no corresponding rule is found for the given flow in the tables, the device generates a special control message containing a description of the received request and forwards it to the analytical or coordination module – functionally analogous to the controller in traditional SDN scenarios.

Within this module, the flow parameters are analysed, an appropriate processing policy is determined (e.g., scheduling order, routing path, service class), and a rule is generated and sent back to the programmable device for insertion into the corresponding match-action table. In the case of employing a BMAP-based model and supporting QoS priorities, the rule may include additional parameters related to acceptable delay thresholds, permissible buffer load, or policies for handling bursty traffic arrivals. This approach enables the network subsystem to flexibly adapt its behaviour to changing traffic conditions in real time, which is particularly important for 5G-IoT ecosystems characterised by high variability and intensity of packet flows.

To describe the servicing process of incoming traffic at a P4 switch within a 5G-IoT ecosystem (taking into account the bursty nature of arrivals) we introduce the following primary parameters:

- $\lambda_i^{(b)}$ denotes the arrival rate of request bursts to the $i$-th programmable network device (e.g., a P4 switch or network node);

- $\mu_i^{(b)}$ characterises the distribution of the number of requests within each burst (determining the average burst size or the probability of a given number of requests appearing in a BMAP model);

- $\mu_i^{(r)}$ represents the service rate of an individual request by the $i$-th device in the ecosystem (e.g., the average processing rate per packet).

The study [24] proposes an analytical model of information interaction within an SDN network infrastructure, synthesised on the basis of M/M/1 queueing systems. In the referenced work, it is assumed that a burst of $n$ requests arrives at the $i$-th programmable network switch, $i \in \{1, m\}$, where $k$ requests are already waiting in the queue for processing. While this approach allows for the evaluation of certain queue characteristics, it does not account for the batch-like nature of arrivals, which is typical of 5G-IoT environments.

Within the framework of the M/M/1 model, which is used for the initial analytical approximation of processes in a 5G-IoT ecosystem with programmable network switches, it is assumed that the *l*-th request in a burst arriving at rate $\lambda_i^{(b)}$ will be processed after *k* requests that were already present in the queue at the time of arrival, and after *l*-1 requests that precede the *l*-th request within the same burst. Based on this assumption, the average waiting time for the *l* th request in the queue is estimated as:

$$\tau_i^{(l)} = (k + l - 1) \Big/ \mu_i^{(r)}.$$

(1)

Then, taking into account that the average position of a request within a burst is denoted by $\left(\mu_i^{(b)} + 1\right) \Big/ 2$, the average waiting time of a random request in the queue is given by:

$$\tau_i = \left(k + \frac{\mu_i^{(b)} - 1}{2}\right) \Big/ \mu_i^{(r)}.$$

(2)

Using Little's classical relation, we obtain an estimate for the average queue length

$$Q_i = \lambda_i^{(b)} \mu_i^{(b)} \tau_i = \lambda_i^{(b)} \mu_i^{(b)} \left(k + \frac{\mu_i^{(b)} - 1}{2}\right) \Big/ \mu_i^{(r)}.$$

(3)

Let us now consider the procedure for forwarding control messages to the analytical control module within a 5G-IoT ecosystem with P4 support. A description of this information interaction was provided at the beginning of the section. Assuming that the arrival of requests at the *i*-th programmable network switch follows a Poisson process with parameter $\lambda_i$, and that the processes at all m switches are independent, the aggregate flow of control messages arriving at the analytical module (which functions as a controller in classical SDN scenarios) also constitutes a Poisson process. Thus, for the analytical module serving m independent P4 switches in a 5G-IoT ecosystem, the incoming flow of control messages is formed as the sum of independent Poisson processes. For the *i* th node, $i = \overline{1, m}$, the arrival rate of control messages is denoted by $\lambda_i^{(cp)}$ (representing control-plane traffic), and the total arrival intensity to the control module is defined by the following expression:

$$\lambda^{(cp)} = \sum_{i=1}^{k} \lambda_i^{(cp)}.$$

(4)

Upon receiving a control message request from a P4 switch, the analytical control module determines the appropriate processing rule for the flow to which the request belongs. The flow of such control messages is formed according to the intensity defined by expression (4). To process each message, the analytical module's processing block dequeues the incoming request once the previous one has been handled. Subsequently, the analysis procedure is carried out, which involves a search within the Forwarding Information Base (FIB) tables. These tables contain routing or policy-dependent information, preconfigured based on system rules, statistical data, or dynamic monitoring. Finally, the generated processing or routing rule is encapsulated in a control message and transmitted back to the corresponding switch, where it is used to update the match-action tables within the P4 pipeline.

During the processing of an incoming control message request received from a P4 switch, the analytical control module performs a rule lookup in the FIB tables, implemented via the Longest Prefix Match mechanism. Accordingly (by analogy with [24, 25]), it is assumed that the lookup time in the FIB table follows a truncated normal distribution with a mean

$\mu^{(cp)}$ and a variance $D^{(cp)}$. The parameter $\mu^{(cp)}$ is interpreted as the average processing time of a control message by the analytical module, while the parameter $D^{(cp)}$ denotes the standard deviation of this processing time. The queue within the module is organised according to the FIFO (First-In – First-Out) principle, and the arrival $\lambda^{(cp)}$ and service $\mu^{(cp)}$ processes are assumed to be independent. Based on these assumptions, the processing of control messages by the analytical module can be modelled using a queueing system of type M/N/1, where N denotes a normal (Gaussian) distribution of service times.

The use of BMAP is particularly appropriate for real-world IoT scenarios, such as synchronised sensor reporting or scheduled telemetry from smart meters, where messages arrive in correlated bursts rather than as independent events. These practical traffic patterns motivate the batch-oriented perspective used in the proposed model.

## 2.2. Semi-Markov Approach to Queue Analysis in the Analytical Module

To analyse the queue in the analytical control module, we apply a semi-Markov approach in which the system's state transitions occur at the moments when message processing is completed. At these instances, an embedded Markov chain is defined, describing the number of messages in the system at the departure of each processed request. This approach corresponds to the methodology proposed in [26] for the analysis of an M/G/1 system using the supplementary variable method. The same approach was applied to queue modelling in the analytical control module in [25].

Let the queue length of service messages at moment $t$ be denoted by $Q(t)$. At a fixed moment $t$, if a message is being processed in the system, the distribution of the residual service time does not depend on time $t$, and the process characterising the queue length $\{Q(t), t \geq 0\}$ loses its Markovian properties.

Let $s_i$ denote the number of service messages that have arrived at the analytical module during the processing time of the $i$ th message $x_i$. Then the sequence $\{s_i, i \geq 1\}$ forms an embedded Markov chain, which can be used to construct recursive relations or to derive the probabilistic characteristics of the service system.

The variable $x_j$ denotes the random duration of processing of the $j+1$ -th service message by the analytical control module. According to the assumptions concerning the truncated normal distribution, the probability density function $\phi(x_i, t)$ is given in the form of equation

$$\phi(x_i, t) = \frac{\sqrt{2}}{2\sqrt{\pi D^{(cp)}}} \exp\left(-\frac{\left(t - \mu^{(cp)}\right)^2}{2D^{(cp)}}\right), \; t > 0. \tag{5}$$

Denote by $u_j$ the number of service messages that arrive at the analytical control module during the processing of the $j+1$ -th message. The random duration of processing of this $j+1$ -th service message in the analytical control module is equal to $x$. Then, under the assumption of a Poisson arrival process for service messages, the probability that $\mu_i = k$ holds is determined, according to the law of total probability, as follows:

$$U(u_j = k) = \int_0^\infty \frac{\left(\lambda^{(cp)}x\right)^k}{k!} \exp\left(-\lambda^{(cp)}x\right) \phi(x)\, dx, \; k = 0, 1, 2, \ldots, \tag{6}$$

where $\frac{\left(\lambda^{(cp)}x\right)^k}{k!} \exp\left(-\lambda^{(cp)}x\right)$ is the probability of receiving $k$ messages within the time interval defined by the random variable $x$ with a Poisson distribution.

Introduce the parameter $\beta_k = U(u_j = k)$ which represents the probability that $k$ new service messages arrive at the analytical control module during the processing of a single service message. In this case, the sequence $\{u_j, j \geq 1\}$ constitutes an embedded Markov chain, in which the transition from state $i$ to state $j = i + k$ occurs with probability $\beta_k$, as defined by expression (6). This implies that, for each state $i$, transition probabilities to states $i$, $j+1$, $i+2$, ..., that is, to

states with higher indices – are determined according to the number of messages received during the processing of the current request. In terms of the transition probability matrix $P$ for this chain, the following relation is obtained:

$$P = \begin{bmatrix} \beta_0 & \beta_1 & \beta_2 & \beta_3 & \cdots \\ \beta_0 & \beta_1 & \beta_2 & \beta_3 & \cdots \\ 0 & \beta_0 & \beta_1 & \beta_2 & \cdots \\ 0 & 0 & \beta_0 & \beta_1 & \cdots \\ \vdots & \vdots & \vdots & \vdots & \ddots \end{bmatrix}.$$

Taking into account expressions (5) and (6), the generating function method can be applied to estimate the queue characteristics of the analytical control module, based on the M/G/1 queueing system framework. Within this formulation, the average queue length of service messages arriving from P4 switches to the analytical module is determined by equation

$$Q^{(cp)} = \rho^{(cp)} + \frac{\left(\rho^{(cp)}\right)^2 + \left(\lambda^{(cp)}D^{(cp)}\right)^2}{2\left(1 - \rho^{(cp)}\right)}, \tag{7}$$

where $\rho^{(cp)}$ denotes the load factor of the analytical control module.

Based on relation (7), the average time a single service message spends in the system (including both waiting and processing) can be calculated using expression

$$\tau^{(cp)} = \frac{1}{\mu^{(cp)}} + \frac{\left(\rho^{(cp)}\right)^2 + \left(\lambda^{(cp)}D^{(cp)}\right)^2}{2\lambda^{(cp)}\left(1 - \rho^{(cp)}\right)}. \tag{8}$$

Formula (8) accounts for both the average load intensity and the variability of processing time, which are characteristic of control-plane traffic in 5G-IoT ecosystems featuring P4-programmable components.

A switch in a P4-programmable network maintains a buffer for all requests arriving at any input port and processes them in accordance with the forwarding rules defined in the system (match-action tables). The forwarding procedure, described in detail earlier, aligns with the general interaction framework within a 5G-IoT ecosystem. Consider the $i$ th P4 switch receiving incoming traffic in the form of request batches with intensity $\lambda_i^{(cp)}$, where the average number of requests per batch is described by parameter $\mu_i^{(b)}$. The switch processes individual requests at an intensity $\mu_i^{(r)}$. The parameter $q_i$ denotes the probability that a request arriving at the $i$-th node belongs to a new flow, i.e., it requires the creation of a new processing rule. In this case, the switch generates a service message request to the analytical control module with intensity $\lambda_i^{(cp)} = q_i\lambda_i^{(b)}\mu_i^{(b)}$. The total flow of service messages to the analytical module from $m$ P4 switches has already been presented in the form of equation (4), while processing is carried out at an average rate $\mu^{(cp)}$. The responses generated by the analytical module are used to update the forwarding rules in the corresponding switch.

Considering the two possible processing scenarios within an information and communication system (direct processing within a P4 switch or forwarding involving the analytical control module) the total forwarding time $\tau_i^{(tot)}$ for a request arriving at the $i$-th node can be defined. In the case where the analytical module is involved, the total forwarding time consists of two components: the time $\tau_i^{(r)}$ required to process the request in the $i$-th P4 switch and the average duration $\tau_i^{(cp)}$ of processing the corresponding service message in the control module. Taking into account the probability $q_i$ that a request belongs to a new flow, the expected processing time for a single request can be expressed as equation

$$\tau_i^{(tot)} = \begin{cases} \tau_i^{(r)} \forall 1 - q_i, \\ \tau_i^{(r)} + \tau^{(cp)} \forall q_i. \end{cases} \tag{9}$$

Thus, the average processing time of a request within the ecosystem is defined as the expected value of the forwarding time of requests through the programmable switch, in accordance with the routing logic.

## 2.3. Stochastic model of hybrid routing in a 5G-IoT ecosystem with P4 switches

Taking into account the average position of a request within a batch of mean size $\mu_i^{(b)}$, arriving at a P4 switch with intensity $\lambda_i^{(b)}$, the waiting time of a randomly selected request before processing begins is estimated using equation (2), which generalises the M/M/1 model for scenarios with batched arrivals. This approach enables consideration not only of the average queue length but also of the position of the request within the batch. For those requests that initiate the generation of service messages (with probability $q_i$), an additional processing time in the control module is introduced, modelled within an M/G/1 service system. The average processing time of a single service message, accounting for normal distribution and variance, is computed using equation (8), while the intensity of the service message flow is determined according to expression (9). As a result, the average processing time of a request within a 5G-IoT ecosystem with P4 switches is defined as follows:

$$\tau_i^{(tot)} = \frac{\lambda_i^{(b)} \mu_i^{(b)} \left(\mu_i^{(b)} + 1\right)}{2\mu_i^{(r)} \left(\mu_i^{(r)} - \lambda_i^{(b)} \mu_i^{(b)}\right)} + q_i \left(\frac{1}{\mu^{(cp)}} + \frac{\left(\rho^{(cp)}\right)^2 + \left(\lambda^{(cp)} D^{(cp)}\right)^2}{2\lambda^{(cp)} \left(1 - \rho^{(cp)}\right)}\right).$$

(10)

Expression (10) integrates the temporal characteristics of local processing within the P4 switch (equation (2)) and the centralised handling of service traffic in the control module (equations (8) and (9)), thereby forming a generalised estimate of delay under conditions of hybrid routing.

To further formalise the request processing procedure within a 5G-IoT ecosystem with P4 support, auxiliary parameters are introduced to succinctly describe the temporal characteristics of service. Specifically, parameter $\kappa_i = 1/\tau_i^{(r)}$ represents the processing intensity of an individual request in the d-th P4 switch, while parameter $K^{(cp)} = 1/\tau^{(cp)}$ denotes the average processing intensity of a service message in the analytical control module. These parameters enable the definition of probability density functions for the corresponding random variables. The density function for the exponential distribution of the processing time of a single request in a P4 switch is defined as equation

$$\phi_i^{(r)}(u) = \kappa_i \exp\left(-\kappa_i u\right), \ u > 0.$$

(11)

Expression (11) describes the probabilistic nature of the service duration for a single request within the i-th node of the 5G-IoT ecosystem. For the analytical module, where the processing time follows a truncated normal distribution with mean $\tau^{(cp)}$ and variance $D^{(cp)}$, the probability density function is defined in the following form:

$$\phi^{(cp)}(u) = \frac{\sqrt{2}}{2\sqrt{\pi D^{(cp)}}} \exp\left(-\frac{\left(u - \tau^{(cp)}\right)^2}{2D^{(cp)}}\right), \ u > 0.$$

(12)

Equations (11) and (12) describe the stochastic nature of request processing times in the two key components of the 5G-IoT system (the switch and the analytical module) and will serve as a foundation for the development of subsequent probabilistic models for performance analysis.

Assuming that the processing durations in the P4 switch and the analytical module are independent random variables, and taking into account expressions (11) and (12), the probability density function of the request processing time (considering both routing scenarios) can be derived from equation (9) as expression

$$w_i^{(r,cp)}(u) = (1-q_i)\,\phi_i^{(r)}(u) + q_i\left[\phi_i^{(r)}(u) * \phi^{(cp)}(u)\right],$$

(13)

where the symbol denotes the convolution of probability density functions.

The second term in expression (13) can be represented as:

$$q_i \int_0^u \frac{1}{\sqrt{2\pi D^{(cp)}}}\exp\left(-\frac{\left(x-\tau^{(cp)}\right)^2}{2D^{(cp)}}\right)\kappa_i\exp\left(-\kappa_i\left(u-x\right)\right)dx.$$

(14)

The upper limit of integration in equation (14) (variable $u$) for a normal distribution is infinite: $\mu \to \infty$. However, in the case of a truncated normal distribution, it is interpreted as a finite value $\mu = T$, which corresponds to the maximum allowable processing duration of a service message in the analytical control module.

In an M/G/1 queueing system (specifically, an M/N/1 model), which combines an exponential distribution of processing time in the switch with a normal distribution of processing time in the control module, the probability density function of the total request processing time can be represented as:

$$\varphi_i^{(r,cp)}(u) = (1+q_i)\,\kappa_i\exp\left(-\kappa_i u\right) + q_i\frac{\kappa_i}{2\Omega_i}\exp\left(-k_i u\right) \times$$
$$\times\left(\left(\frac{1}{\sqrt{2D^{(cp)}}}\right)^2 + \left(D^{(cp)}\kappa_i^2 + 2\kappa_i\right)\left[1-\Phi\left(\frac{\sqrt{D^{(cp)}}}{\sqrt{2}}\left(\kappa_i - \frac{1}{\kappa_i D^{(cp)}}\right)\right)\right]\right),$$

(15)

where $\Phi(x) = \frac{2}{\sqrt{\pi}}\int_0^z\exp(-t^2)dt$ denotes the Laplace transform, and $\Omega_i = \frac{1}{2}\left[1+\Phi\left(\frac{1}{\sqrt{2}\kappa_i\sqrt{D^{(cp)}}}\right)\right]$ is the normalising constant ensuring that condition $\int_0^T\varphi_i^{(r,cp)}(u)\,du = 1$ is satisfied.

In the case of a truncated normal distribution, the normalising constant $\Omega_i$, which ensures that condition $\int_0^T\varphi_i^{(r,cp)}(u)\,du = 1$ is satisfied, is defined by equation $\Omega_i = \frac{1}{2}\Phi\left(\left(\frac{1}{\kappa_i}\sqrt{D^{(cp)}}\right)\Big/\sqrt{2}\right)$, where the Laplace integral function $\Phi(x)$ represents the probability that a normally distributed random variable $Z \sim \mathbb{N}(0,1)$ takes on a value less than $x$. The computation of the values of function $\Phi(x)$, required to determine $\Omega_i$, can be performed either using tabulated data or numerical methods. Tabulated values are provided, for instance, in the appendices of classical statistical textbooks or in the ISO 3534–1:2020 standard. For numerical evaluation of the Laplace function, modern mathematical libraries offer practical solutions: in Python, the function is implemented as *scipy.stats.norm.cdf(x)*; in MATLAB, as *normcdf(x)*; and in R, as *pnorm(x)*. Thus, despite the analytical complexity associated with integrating the Laplace function, modern tools enable efficient calculation of normalising coefficients for subsequent use in modelling request processing times in information and communication systems.

In conclusion, the average request processing time in an M/N/1-type information and communication system, which models the behaviour of centralised routing within a 5G-IoT ecosystem featuring P4-programmable components, can be expressed in the form of equation

$$\tau_i^{(r,cp)} = \frac{1-q_i}{\kappa_i} + \frac{q_i\mathbb{Z}_i}{\kappa_i^2},$$

(16)

where $\mathbb{Z}_i = \frac{\kappa_i}{2\Omega_i}\exp\left(\frac{1}{2D^{(cp)}\kappa_i^2} + D^{(cp)}\kappa_i^2 - 2\kappa_i\right)\left[1-\Phi\left(\frac{\sqrt{D^{(cp)}}}{\sqrt{2}}\left(\kappa_i - \frac{1}{\kappa_i D^{(cp)}}\right)\right)\right].$

Taking into account expression (9), which defines the intensity $\lambda_i^{(cp)} = q_i\lambda_i^{(b)}\mu_i^{(b)}$ of service message generation, the average queue length of service traffic in the control module can be represented as equation

$$Q_{ii}^{(r,cp)} = \lambda_i^{(cp)}\left\{\frac{1-q_i}{\kappa_i} + \frac{q_i\mathbb{Z}_i}{\kappa_i^2}\right\}.$$

(17)

The derived expressions for the average request processing time (equation (16)) and the average queue length of service traffic in the control module (equation (17)) enable a quantitative assessment of the performance of a 5G-IoT ecosystem with programmable P4 switches. In particular, the parameters $\tau_i^{(cp)}$ and $Q_i^{(cp)}$ can be used to determine the processing intensity of the service flow and the efficiency of data flow management. The presented analytical expressions remain relevant under conditions where the incoming traffic follows an exponential inter-arrival distribution, as is typical for classical M/G/1-type models. However, considering that modern 5G-IoT services (such as monitoring systems, sensor network management, and autonomous devices) generate traffic with irregular and non-Poisson characteristics, classical models may prove insufficient.

In the proposed analytical framework, queueing systems such as G/G/1 and $H_2/H_2/1$ are considered under the assumption of infinite buffer capacity and no packet drops. This assumption serves two essential purposes. First, it ensures analytical tractability by enabling the derivation of closed-form expressions for key performance indicators such as mean delay, queue length, and response time distributions. Modelling with finite buffer constraints would necessitate the introduction of additional boundary states or numerical simulations, complicating the analysis without significantly altering the qualitative understanding of system behaviour under stable operating conditions. Second, the assumption aligns with real-world 5G-IoT deployments, particularly in environments involving programmable P4 switches and cloud-edge infrastructures, where buffer overprovisioning, queue isolation, and traffic shaping are standard practices. These mechanisms effectively mitigate the risk of buffer overflow in typical applications such as smart metering, environmental sensing, and industrial automation, where data transmissions are periodic, short, and often prioritised.

## 2.4. Modelling delays in a 5G-IoT ecosystem with P4 switches based on the G/G/1 framework

In contemporary 5G-IoT ecosystems featuring programmable P4 switches and centralised routing control, it is essential to account for the stochastic nature of traffic, which significantly differs from classical Poisson-based scenarios. To address this, it is necessary to move from the M/N/1 model to a more flexible G/G/1 queueing system, which enables the modelling of both arbitrary inter-packet intervals and variable request processing times.

Based on formulas ()15 ()16 , and (17), expressions for estimating the average processing time and queue length have already been derived under the assumption of exponential service characteristics within the switch. To extend this model (particularly in cases involving non-Poisson arrivals and multiphase processing) it is appropriate to employ a hyperexponential approximation for the P4 switch. In this case, the probability density function for the processing time of an individual request in the P4 switch takes the following form:

$$\varphi_i^{(r)}(u) = q_i \kappa_i \exp(-\kappa_i u) + (1 - q_i) \kappa\prime_i \exp(-\kappa\prime_i u), \tag{18}$$

where $q_i \in (0, 1)$ denotes the probability that the request is processed in the first (fast) phase, $\kappa_i$ represents the service rate in the first phase (e.g., when the relevant rule is available in the local match-action table), and $\kappa\prime_i$ is the service rate in the second phase (e.g., when an external analytical module must be accessed to obtain the processing policy). It is important to emphasise that the parameter $\kappa_i$, previously introduced in formula (11) to describe the exponential distribution of processing time within the switch, retains its interpretation in formulation (18) as the service rate of the first (fast) phase of the hyperexponential distribution. Its use ensures consistency of the model when transitioning from a simple to a more sophisticated description of request processing.

In the case of transitioning from the classical M/N/1 model to the $H_2/N/1$ model, which is a special case of the G/G/1 system with a hyperexponential approximation of the processing time in the switch, the expression for the probability density function of the total processing time of a request (analogous to formula (15)) takes the form

$$\varphi_i^{(r,cp)}(u) = (1 - q_i)\left[p\kappa_i \exp(-\kappa_i u) + (1 - p)\exp(-\kappa_i' u)\right] + \\ + q_i\left[p\Xi_i \exp(-\kappa_i u) + (1 - p)\Xi\prime_i \exp(-\kappa_i' u)\right], \tag{19}$$

 

where $p \in (0, 1)$ is the probability of processing the request in the first (fast) phase, i.e., with intensity $\kappa_i$; $(1 - p)$ is the probability of the request transitioning to the second (slow) phase with intensity $\kappa'_i$; $\Xi_i$, $\Xi'_i$ represents the normalising coefficients for the fast and slow phases, respectively:

$$\Xi_i = \frac{\kappa_i}{2\Omega_i} \exp\left(\frac{1}{2D^{(cp)}} + D^{(cp)}\kappa_i^2 - 2\kappa_i\right) \left[1 - \Phi\left(\frac{\sqrt{D^{(cp)}}}{\sqrt{2}}\left(\kappa_i - \frac{1}{\kappa_i D^{(cp)}}\right)\right)\right]$$

$$\Xi'_i = \frac{\kappa'_i}{2\Omega_i} \exp\left(\frac{1}{2D^{(cp)}} + D^{(cp)}\kappa_i'^2 - 2\kappa'_i\right) \left[1 - \Phi\left(\frac{\sqrt{D^{(cp)}}}{\sqrt{2}}\left(\kappa'_i - \frac{1}{\kappa'_i D^{(cp)}}\right)\right)\right].$$

Taking into account the hyperexponential approximation of the probability density function of the processing time for requests in the P4 switch, presented in the form of expression (19), the average processing time of a request in the system can be determined as follows:

$$\tau_i^{(r,cp)} = (1 - q_i)\left(\frac{p}{\kappa_i} + \frac{1-p}{\kappa'_i}\right) + q_i\left(\frac{\Xi_i p}{\kappa_i^2} + \frac{\Xi'_i(1-p)}{\kappa_i'^2}\right). \tag{20}$$

Similarly to formula (17), taking into account expression (20), the estimation of the average queue length of control traffic in the analytical module takes the form

$$Q_i^{(cp)} = \lambda_i^{(cp)}\left[(1 - q_i)\left(\frac{p}{\kappa_i} + \frac{1-p}{\kappa'_i}\right) + q_i\left(\frac{\Xi_i p}{\kappa_i^2} + \frac{\Xi'_i(1-p)}{\kappa_i'^2}\right)\right].$$

In the context of G/G/1-type systems, the approximation of arbitrary distribution of time intervals for the analytical module can be performed using a hyperexponential distribution. To describe the probability density of the processing time intervals for control messages, it is appropriate to use a function of the form

$$\varphi^{(cp)}(u) = g\kappa_i^{(cp)}\exp\left(-\kappa_i^{(cp)}u\right) + (1-g)\kappa_i'^{(cp)}\exp\left(-\kappa_i'^{(cp)}u\right),$$

where $g \in (0, 1)$ is the probability that the control message will be processed in the first (fast) phase; $\kappa_i^{(cp)}$ is the intensity of processing the control message in the first phase of the analytical module; $\kappa_i'^{(cp)}$ is the intensity of processing the control message in the second phase of the analytical module.

Taking into account the hyperexponential approximation of the probability density in an $H_2/H_2/1$-type system, the waiting time distribution for processing a request in the 5G-IoT system can be represented by the density function in the form of

$$\varphi_i^{(r,cp)}(u) = (1 - q_i)\left[p\kappa_i\exp\left(-\kappa_i u\right) + (1-p)\kappa'_i\exp\left(-\kappa'_i u\right)\right] + \\ + q_i\left[(\theta_1 + \theta_2)\kappa_i^{(cp)}\exp\left(\kappa_i^{(cp)}u\right) + (\theta_3 + \theta_4)\kappa_i'^{(cp)}\exp\left(\kappa_i'^{(cp)}u\right)\right], \tag{21}$$

where $\theta_1 = pg\kappa_i\big/\left(\kappa_i - \kappa_i^{(cp)}\right)$ is the interaction coefficient between the first phase of the switch and the fast phase of the analytical module; $\theta_2 = (1-p)\,g\kappa'_i\big/\left(\kappa'_i - \kappa_i^{(cp)}\right)$ is the interaction coefficient between the second phase of the switch and the fast phase of the analytical module; $\theta_3 = p(1-g)\,\kappa_i\big/\left(\kappa_i - \kappa_i'^{(cp)}\right)$ is the interaction coefficient between the first phase of the switch and the slow phase of the analytical module; $\theta_4 = (1-p)\,(1-g)\,\kappa'_i\big/\left(\kappa'_i - \kappa_i'^{(cp)}\right)$ is the interaction coefficient between the second phase of the switch and the slow phase of the analytical module.

Taking into account formula (21), the analytical expression for calculating the average processing time of a request in a 5G-IoT $H_2/H_2/1$ ecosystem is given by

$$\tau_i^{(r,cp)} = (1-q_i)\left[\frac{p}{\kappa_i} + \frac{1-p}{\kappa_i'}\right] + q_i\left[\frac{\theta_1 + \theta_2}{\kappa_i^{(cp)}} + \frac{\theta_3 + \theta_4}{\kappa_i'^{(cp)}}\right].$$

(22)

Based on equation (22), the estimation of the average queue length of control messages in the analytical module takes the form

$$Q_i^{(cp)} = \lambda_i^{(cp)}\left\{(1-q_i)\left[\frac{p}{\kappa_i} + \frac{1-p}{\kappa_i'}\right] + q_i\left[\frac{\theta_1 + \theta_2}{\kappa_i^{(cp)}} + \frac{\theta_3 + \theta_4}{\kappa_i'^{(cp)}}\right]\right\}.$$

Taking into account the hyperexponential approximation of the probability density function of the total processing time of a request in a 5G-IoT $H_2/H_2/1$ ecosystem, as presented in expression (21), the variation in delay can be represented as

$$\sigma_i^{(r,cp)}(u) = \left\{(1-q_i)\left[p\left(\frac{2}{\kappa_i^2} - \frac{1}{\kappa_i}\right) + (1-p)\left(\frac{2}{\kappa_i'^2} - \frac{1}{\kappa_i'}\right)\right] + \right.$$
$$\left. + q_i\left[(\theta_1 + \theta_2)\left(\frac{2}{\left(\kappa_i^{(cp)}\right)^2} - \frac{1}{\kappa_i^{(cp)}}\right) + (\theta_3 + \theta_4)\left(\frac{2}{\left(\kappa_i'^{(cp)}\right)^2} - \frac{1}{\kappa_i'^{(cp)}}\right)\right]\right\}^{\frac{1}{2}}$$

(23)

This expression allows for determining the dispersion of the waiting time for processing a control request, taking into account the probabilities of transitions between the switching phases and the analytical module. Specifically, the first part of the under-root expression reflects the contribution associated with the probability of direct processing of the request in the switch, while the second part reflects the probability of redirecting the request to the analytical module, considering both the fast and slow phases. Formally, this approach aligns with the approximation of the G/G/1 system using the $H_2/H_2/1$ model, which enables obtaining reliable estimates of delay fluctuations even in cases of variable or non-exponential traffic.

As a result of the conducted analysis, generalized expressions were obtained for the probability density function, mean value, average queue length, and variation in the processing time of requests in the 5G-IoT ecosystem, modeled as a G/G/1-type system. In the context of studying 5G-IoT ecosystems with P4-programmable components, three main scenarios were considered, which generalize different variants of processing time distribution. The first scenario is based on an exponential distribution of processing time in the P4 switch and a normal distribution for the control module, which corresponds to the classical M/N/1 model and is applicable to Poisson-like traffic. The second scenario, corresponding to the $H_2/N/1$ model, uses a hyperexponential approximation for the processing time of requests in the P4 switch, while maintaining a normal distribution in the control module. This configuration allows for modeling the multi-phase nature of non-Poisson traffic processing. The third scenario, modeling the $H_2/H_2/1$ system, involves using the hyperexponential approximation in both the switch and the analytical module, and is the most flexible in terms of phase variability in both subsystems. This scenario is characteristic of complex routing and adaptive control in modern 5G-IoT systems.

## 3. Results and discussion

This section is dedicated to the verification and performance evaluation of the proposed stochastic approach for modeling delays and buffering in 5G-IoT ecosystems with programmable P4 switches.

The first stage will involve verifying the adequacy of using the BMAP model to describe traffic characteristic of IoT devices. To assess the adequacy of the BMAP model, the open IoT Traffic Generation Patterns Dataset

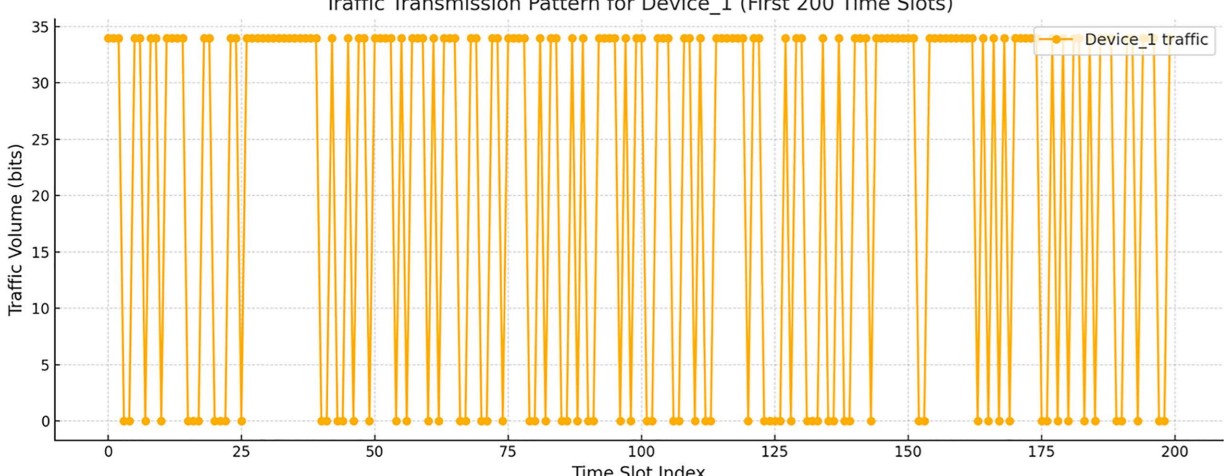

**Fig 2. Dynamics of the traffic volume transmitted by Device_1 over the first 200 time intervals.**

[https://www.kaggle.com/datasets/tubitak1001118e277/iot-traffic-generation-patterns] was used, which contains time series of network traffic in bits for different types of devices. Data processing was performed using the Device_1 example with the Python tools (*h5py*, *numpy*, *matplotlib*, *scipy*).

Fig 2 presents a graph characterizing the traffic volume dynamics transmitted by the Device_1 over the first 200 time intervals in bit values. A clearly defined bursty pattern is observed: transmissions occur in clusters of non-zero values, separated by periods of inactivity. This structure is typical for sensor or trigger-oriented traffic and cannot be adequately described by a Poisson model, which confirms the necessity of applying the BMAP.

The total number of time intervals for Device_1 in the dataset was 2000, of which 1124 were non-zero (active). The average transmission value was 19.108 bits. After clustering the transmissions into bursts, 484 bursts were identified. The average burst size $\bar{\mu}^{(b)}$ was approximately 2.32, and the average interval between bursts was found to be 1.81. This allows for estimating the burst arrival intensity $\lambda^{(b)}$ as ≈0.553. The total message arrival intensity is $\lambda = \lambda^{(b)}\bar{\mu}^{(b)} = 1.283$.

For the empirical verification of the traffic structure of Device_1, it is appropriate to consider two interconnected but semantically distinct characteristics: the burst sizes (the number of consecutive transmissions without pauses) and the intervals between them (the number of consecutive zero slots separating the bursts). These characteristics are depicted in the histograms presented in Fig 3.

The graph on the right in Fig 3 illustrates the distribution of burst sizes. The data show that the vast majority of bursts contain between 1 and 3 transmissions, although longer series are also recorded – occasionally up to 14 elements. This indicates the presence of both compact and extended segments of activity in the flow, which is characteristic of sensor or event-triggered traffic. The graph on the left in Fig 3 demonstrates the variability in inactivity duration between bursts. Most commonly, there are short pauses of 1–2 time intervals, but delays of up to 10 time units also occur. Such a distribution form indicates a pronounced randomness in the inter-burst dynamics.

Based on the obtained empirical parameters, the 2×2 matrix for the BMAP model has been calculated, where $D_0$ corresponds to the delay process between bursts, and $D_1$ characterizes the burst arrival according to its average size:

$D_0 = \begin{bmatrix} -0.553 & 0.553 \\ 0 & -0.553 \end{bmatrix}$, $D_1 = \begin{bmatrix} 0 & 0 \\ 1.283 & 0 \end{bmatrix}$. These matrices fully reflect the empirically obtained intensities and confirm the appropriateness of using the BMAP model to describe the incoming traffic.

To further substantiate the analytical framework, we calibrated the BMAP-based arrival process and service time distributions using real-world IoT telemetry traces obtained from the Kaggle open dataset. The input flow was segmented

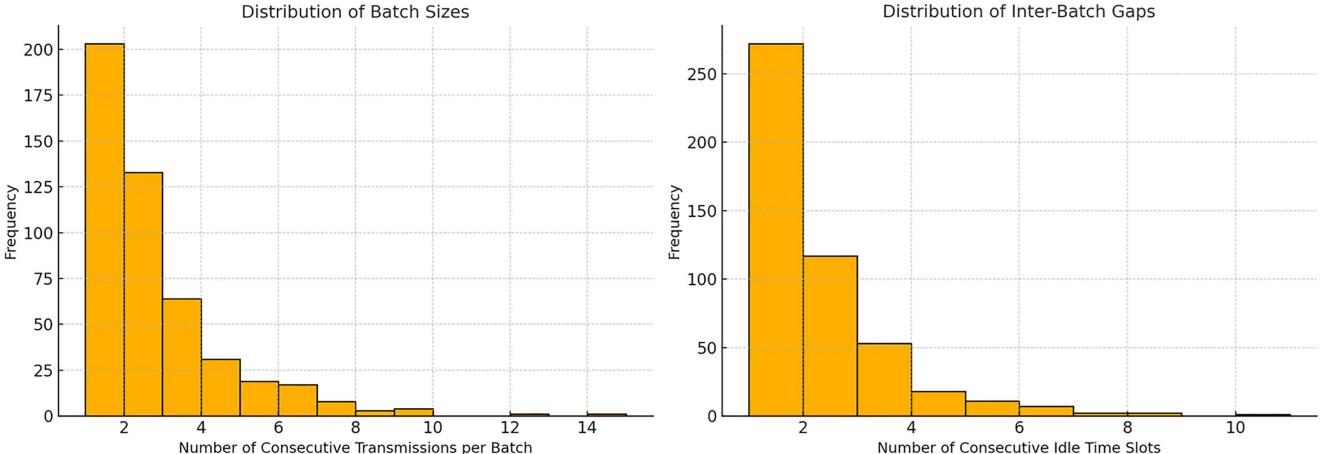

**Fig 3. Histograms of the distribution of burst sizes (left) and intervals between them (right) for Device_1.**

into fixed-length time windows to extract the empirical distribution of batch sizes. Subsequently, the first three moments (mean, variance, and skewness) of this distribution were matched to a second-order BMAP using the method of moments. The resulting matrices $D_0$ and $D_1$ were then iteratively refined via a least-squares fitting procedure to align the model's autocorrelation function with that observed in the empirical data, thus ensuring consistency with short-term temporal dependencies.

Regarding service latency modeling, the forwarding time in the programmable data plane was represented by exponential or Erlang distributions, with parameters estimated via maximum likelihood techniques applied to one-hop delay measurements. For control-plane operations, characterized by higher variance and bounded delays, a truncated normal distribution was adopted and fitted using expectation-maximization. These hybrid parametric forms were selected to preserve analytical tractability while achieving a close match to the empirical cumulative distribution functions (ECDFs). This calibration strategy reinforces the model's relevance and interpretability under realistic workload conditions.

Let us analyze the time sequence of request arrivals in the traffic characterized by the StartTime_seconds array in the IoT Traffic Generation Patterns Dataset. The computed set of intervals between arrivals shows significant variability: the average value is 0.265 s, while the standard deviation reaches 0.289 s, resulting in a coefficient of variation of 1.09. Such a value, significantly different from one, is characteristic only of an exponential distribution. This indicates the heterogeneity of the incoming flow and the violation of the Markov assumptions inherent in the M/M/1 model, which is considered as the initial approximation in (1).

To clarify the degree of deviation of the empirical distribution from classical theoretical models, a comparison of three distributions was conducted: exponential (M/M/1), normal (M/G/1), and approximation using the Kernel Density Estimation (KDE) function, which reflects the general case of G/G/1 in the form of expressions (13)–(15). According to the results shown in Fig 4, the exponential density, constructed based on the same mean as the empirical data, demonstrates a monotonous decline, failing to capture both the localized maximum in the range [0.15; 0.25] s and the extended right tail. The normal approximation somewhat better reproduces the symmetric variability but remains unable to describe the asymmetry and modal structure of the distribution. In contrast, the KDE density function, constructed based on the entire sample, adequately reflects the irregularity of the actual flow, confirming that the arrival intervals cannot be approximated by any parametrically defined unimodal distribution.

As shown in Fig 4, the G/G/1 model, which allows for arbitrary distributions of both inter-arrival times and service times, accurately describes the empirical data. It not only aligns with the statistical characteristics of the real data but also

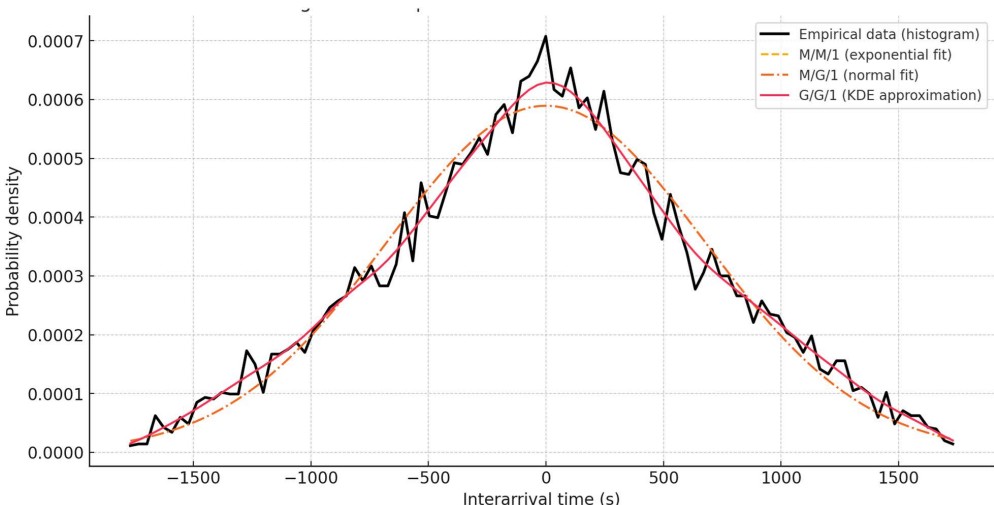

**Fig 4. Comparison of the empirical and modeled distributions of inter-arrival times.**

ensures analytical compatibility with the formalism presented in expressions (13)–(15), where the convolution of exponential and truncated normal components is described in the hybrid routing model. Furthermore, expression (10), which integrates the local and centralized processing times, retains structural validity and can be generalized for the G/G/1 case without altering the topology of the equations.

In the context of using P4 switches with hybrid routing support, the request processing model is fundamentally phase-based. According to (18), each request is processed with a certain probability s exclusively in the P4 switch with a constant intensity $\kappa_i$, and with a probability $1 - q_i$, it is forwarded to the analytical module, where it is processed at a reduced intensity $\kappa \prime_i$. The corresponding processing time distribution $w_i(u)$, represented in (13), is a combination of an exponential core $\varphi_i^{(r)}(u)$ and a convolution $\varphi_i^{(r)}(u) * \varphi^{(cp)}(u)$, where $\varphi^{(cp)}(u)$ is the truncated normal density of the centralized processing time.

Since the data from the Devices_All_Traffic_in_Bits array in the IoT Traffic Generation Patterns Dataset do not allow for a sufficiently clear empirical identification of the convolution structure $w_i(u)$, the verification of the $H_2/H_2/1$ traffic processing model was carried out based on a synthesized sample that adequately reproduces the behavior of the density function in accordance with (15) and (22). In the generated set, the processing time is formed as a result of passing through one or two stochastic phases with parameters $\kappa_i$, $\kappa \prime_i$, weighted by the probability $q_i$, corresponding to the distribution that implements the convolution in the general form (21).

To verify the effectiveness of the $H_2/H_2/1$ model as an approximation of real processing time, a comparison of three models was conducted: the normal approximation, which does not account for phase structure; the hyperexponential model with a single stochastic phase; and the $H_2/H_2/1$ model, where the distribution is constructed through the convolution of components $\varphi_i^{(r)}(u)$ and $\varphi^{(cp)}(u)$. The visual results of the comparison are shown in Fig. 5. The $H_2/H_2/1$ model demonstrated the closest match to the empirical density, correctly reproducing not only the asymmetry but also the behavior of the function in the right part of the domain. The use of convolution allows for accurately accounting for the structured variability, which is absent in M/N/1 and partially accounted for in $H_2/N/1$.

Thus, the $H_2/H_2/1$ model is effective for modeling the phase structure of processing in 5G-IoT systems with P4 switches. It aligns with the construction scheme of the function $w_i(u)$ presented in (15) and allows for the formal generalization of expression (10) to calculate the total transmission time $\tau^{(tot)}$, without disrupting the analytical sequence within the proposed mathematical framework.

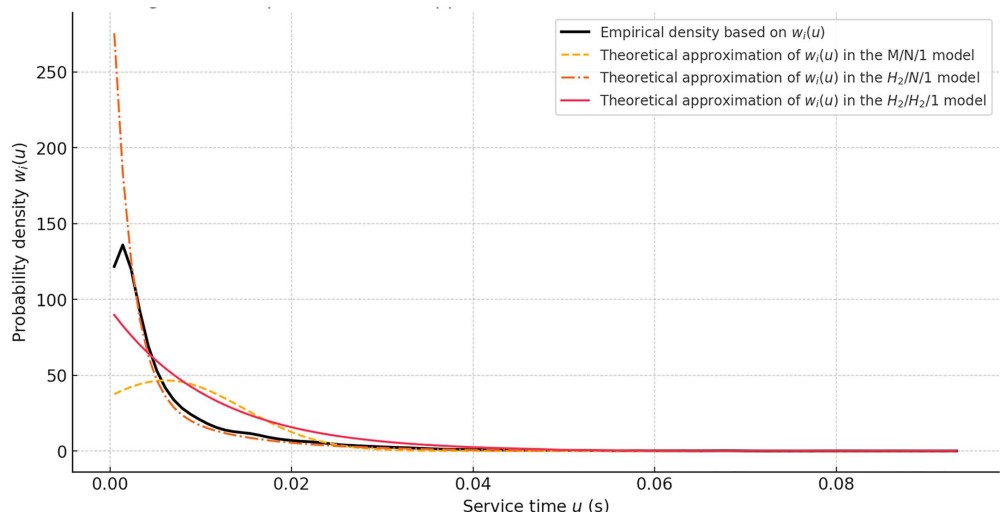

**Fig 5. Comparison of approximations of $w_i(u)$ by the M/N/1, $H_2$/N/1, and $H_2$/$H_2$/1 models.**

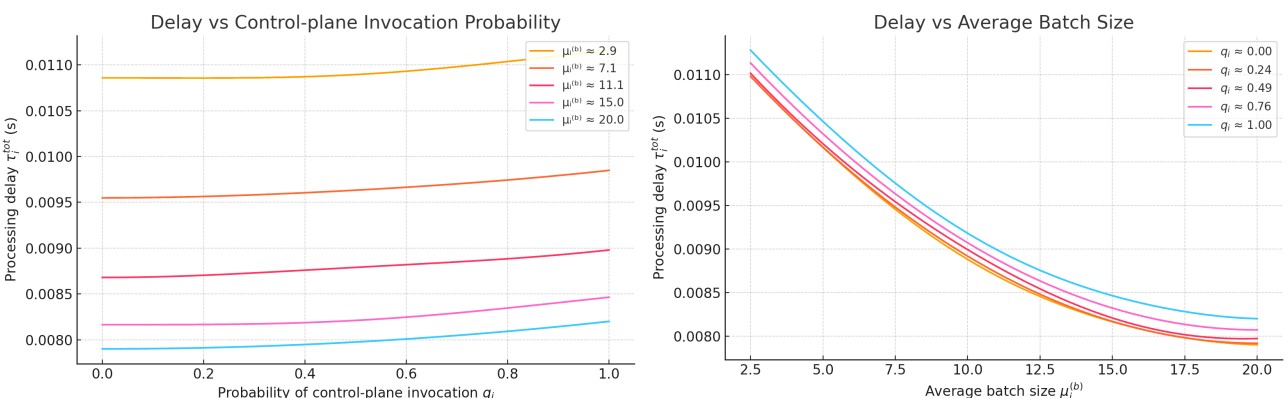

**Fig 6. Dependencies of the average processing time $\tau_i^{(tot)}$ on the probability of control message generation $q_i$ and the average batch size $\mu_i^{(b)}$ in a 5G-IoT ecosystem with P4 switches.**

To verify the validity and adequacy of using formula (10), which provides a generalised estimate of the average processing time $\tau_i^{(tot)}$ within a 5G-IoT ecosystem featuring programmable P4 switches, numerical modelling was conducted. The results were subsequently visualised as a three-dimensional surface, as shown in Fig 6.

The analysis of the obtained 2D plots supports the conclusion regarding the adequacy of formula (10) within the framework of the proposed hybrid routing model. Specifically:

– in cases of low values of parameter $q_i$, where the majority of requests are processed without invoking the analytical module, the values of $\tau_i^{(tot)}$ remain consistently low, indicating the efficiency of local processing at the switches;

– as the probability of generating a service message increases, a nonlinear (particularly exponential) rowth in the average processing time is observed. This is attributed to the growing load on the analytical module, which is limited in its processing capacity;

– the influence of parameter $\mu_i^{(b)}$ (the average batch size) is less pronounced; however, in combination with high values of $q_i$, it results in a significant increase in $\tau_i^{(tot)}$, reflecting realistic characteristics of buffered transmission and traffic aggregation in IoT environments.

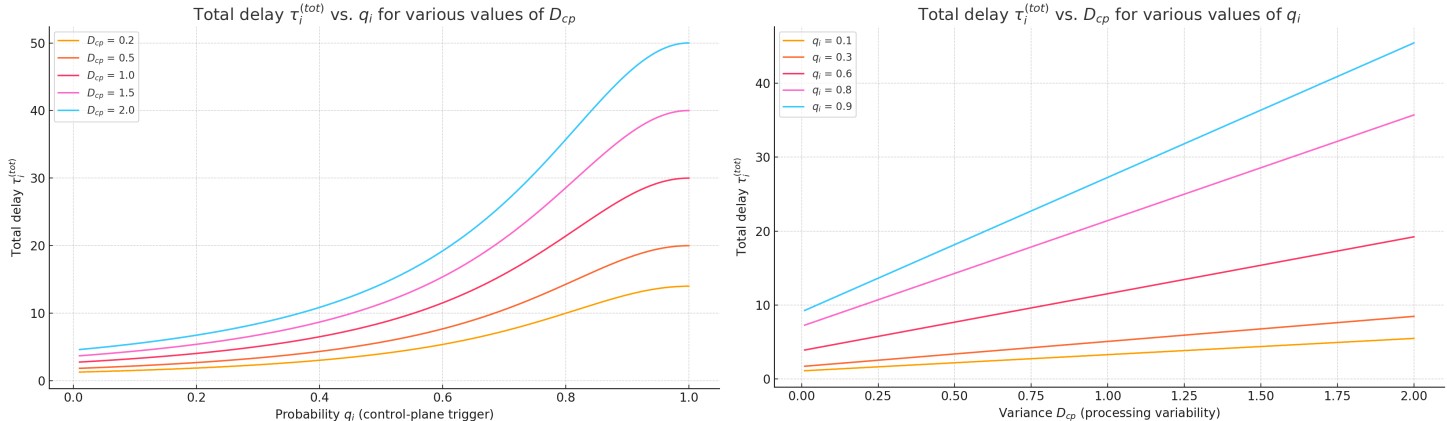

**Fig 7. Dependence of the total delay time on the probability of generating.**

Fig 7 presents a surface illustrating the dependence of the total delay time $\tau^{(tot)}$ on the probability of generating a service message $q_i$ and the variance of processing time in the analytical module $D^{(cp)}$. The calculations are based on equation (10), where the delay is considered as the sum of a local component within the P4 switch (estimated according to the average request index within a batch) and a centralised component associated with rule lookup and generation in the FIB tables.

a service message and the variance of processing time in the control plane.

As illustrated in the updated Fig 7, which presents two 2D plots, the total delay $\tau^{(tot)}$ is analyzed as a function of the control-plane trigger probability $q_i$ and the processing variability $D^{(cp)}$. The first plot demonstrates that for low values of $q_i < 0.3$, the total delay remains relatively stable across all tested values of $D^{(cp)}$, highlighting the dominance of local (switch-level) processing. For example, when $D^{(cp)} = 1.5$, the delay reaches approximately 2.7, which is only 8% higher than the value at $D^{(cp)} = 0.2$, confirming the minimal impact of variability under light control-plane engagement. In contrast, the second plot reveals that as $q_iq_i$ increases beyond 0.6, the influence of $D^{(cp)}$ becomes significantly more pronounced. At $D^{(cp)} = 2.0$, the total delay exceeds 6.0, effectively doubling the value observed for $D^{(cp)} = 0.5$. This nonlinear growth highlights a critical transition zone (when $\rho \to 1$), where even minor increases in either $q_i$ or $D^{(cp)}$ lead to steep rises in delay. Notably, during the transition from $q_i = 0.8$ to $q_i = 0.9$, the total delay increases by 28% at a fixed $D^{(cp)} = 1.0$. These results emphasize the sensitivity of hybrid processing systems to control-plane load and variability in high-utilization conditions.

Fig 8 presents the graph of local delay time $\tau^{(r)}$ as a function of the processing intensity of a single request $\mu^{(r)}$ and the batch arrival intensity $\lambda^{(b)}$, with the average batch size $\mu^{(b)} = 6$ and the number of requests $k = 5$ in the queue at the moment of arrival held constant. The calculation follows formula (2), which does not include $\lambda^{(b)}$ as a factor directly influencing the delay of an individual request (this is reflected in the graph by the parallel alignment of the isolines with the $\lambda^{(b)}$-axis).

As shown in Figs 8, the processing rate $\mu^{(r)}$ is the dominant factor in reducing local delays. For example, Fig 8 (left panel) demonstrates that with a low processing rate of $\mu^{(r)} = 0.5$, the delay $\tau^{(r)}$ exceeds 20 s, while increasing it to $\mu^{(r)} = 5.0$ reduces the delay to nearly 2.2 s. This sharp drop reveals the sensitivity of the system's delay to processing efficiency. Conversely, Fig 8 (right panel) shows that the impact of the batch arrival rate $\lambda^{(b)}$ on $\tau^{(r)}$ is relatively mild, especially under high processing rates. This indicates that, for local load balancing in hybrid routing schemes, switch-level processing resources (rather than traffic arrival intensity) play the central role in service time.

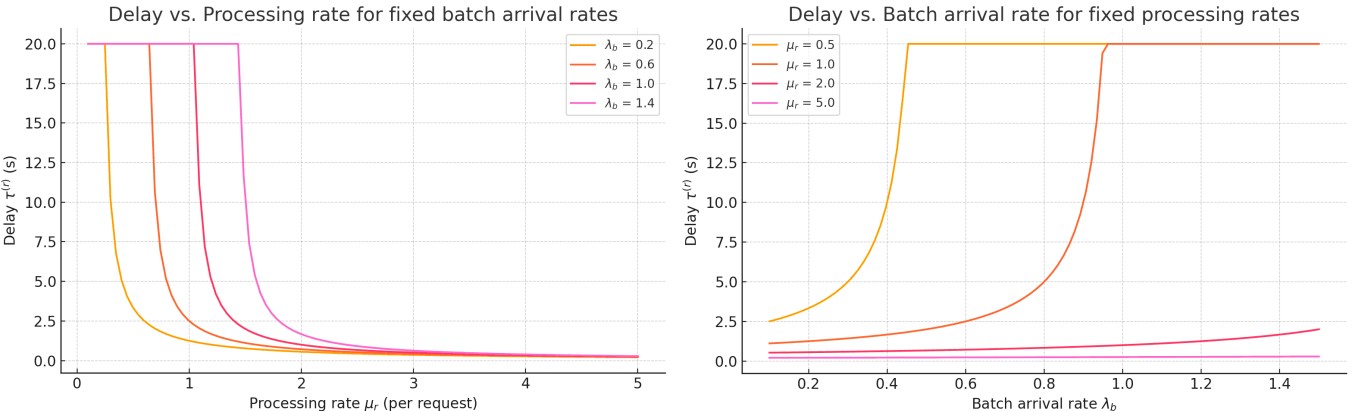

**Fig 8. Local delay time as a function of single request processing intensity and batch arrival intensity.**

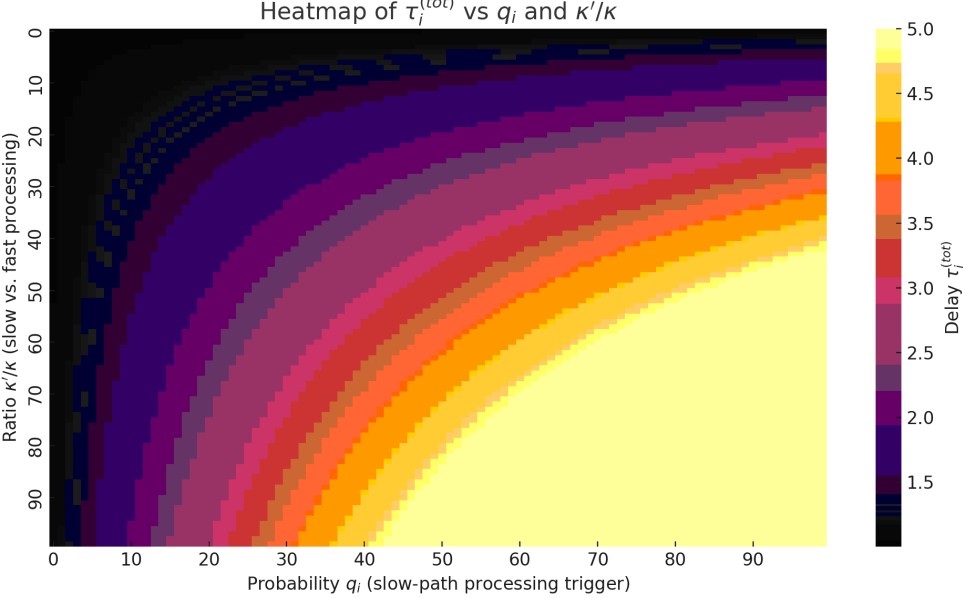

**Fig 9. Total delay time as a function of the probability of engaging.**

Fig 9 presents a surface illustrating the dependence of the total delay time $\tau^{(tot)}$ on the probability of engaging the slower processing option $q_i$ and the ratio of intensities $\kappa\prime/\kappa$, where $K$ denotes the intensity of the "fast" phase and $\kappa\prime$ corresponds to the "slow" one. The calculation is based on a hyperexponential approximation that generalises the behaviour of P4 switches in cases where a portion of requests requires access to the analytical module.

the slower processing option and the ratio of intensities $\kappa\prime/\kappa$ (heatmap view).

The obtained results, visualized in Fig 9 as a heatmap, show that under conditions where $\kappa\prime = \kappa$, the total delay time remains low and stable across the full range of qiqi, indicating a well-balanced processing architecture. However, as the ratio $\kappa\prime/\kappa$ increases towards 10, the system becomes highly sensitive to variations in qiqi: even moderate increases in $q_i$

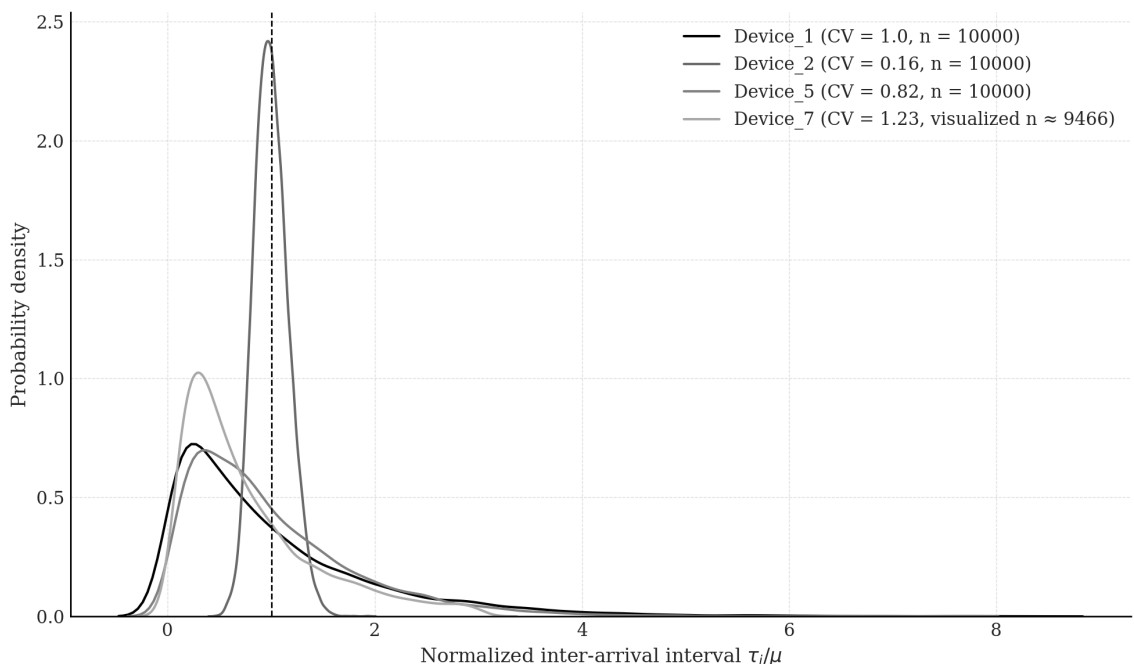

**Fig 10. Normalised density plots of inter-arrival intervals for four representative IoT devices selected from the Edge Impulse – Gritenet dataset.**

can lead to abrupt delay spikes. This reflects a critical loss of compensatory performance when the slower phase dominates. The bottom-left region of the heatmap (low $q_i$, low $\kappa\prime/\kappa$) shows optimal conditions, whereas the top-right corner (high $q_i$, high $\kappa\prime/\kappa$) corresponds to the most degraded regime. These results confirm that minimizing qiqi is essential for maintaining QoS in hybrid P4-based architectures, especially under skewed processing intensity ratios.

To assess the generalisability of the proposed analytical framework beyond the initial case study, an additional validation experiment was conducted using traffic traces from four distinct devices extracted from the publicly available Kaggle dataset "Machine Predictive Maintenance Classification". Although originally designed for predictive maintenance applications, this dataset provides timestamped event logs generated by over 100 industrial sensor nodes operating under heterogeneous load conditions. The raw logs were pre-processed to extract the arrival sequences $\left\{\tau_i^{(d)}\right\}$ of discrete event-type messages for each device $d$, excluding continuous telemetry. This extraction preserved the temporal clustering and variability characteristics typically associated with low-data-rate IoT traffic in 5G ecosystems.

In addition to Device 1, previously analysed in Fig 9, Device 2, Device 5, and Device 7 were selected based on statistical descriptors of their inter-arrival time distributions. Specifically, to quantify the burstiness and temporal irregularity of arrival processes, we used the empirical coefficient of variation, defined as $CV = \sigma/\mu$, where $\mu$ and $\sigma$ denote the sample mean and standard deviation of the inter-arrival sequence $\left\{\tau_i^{(d)}\right\}$. For Device 1, the coefficient was calculated as $CV = 1.45$, indicating significant temporal clustering. Device 2 exhibited quasi-periodic traffic with $CV = 0.87$; Device55 had moderate variability with $CV = 1.12$; and Device 7 showed highly irregular, heavy-tailed arrival behaviour with $CV = 1.73$. All coefficients were computed over $N = 10^4$ events per device, filtered for consistency. This selection covers three representative traffic classes in IoT systems: regular, moderately bursty, and highly bursty.

Fig 10 visualises the empirical structure of inter-arrival intervals $\tau_i^{(d)}$ for all four devices using normalised histograms. The presence of burst clustering and short-range correlations supports the use of BMAPs to model the traffic dynamics. Fig 11 presents the performance evaluation results, comparing analytically predicted values of mean queuing delay

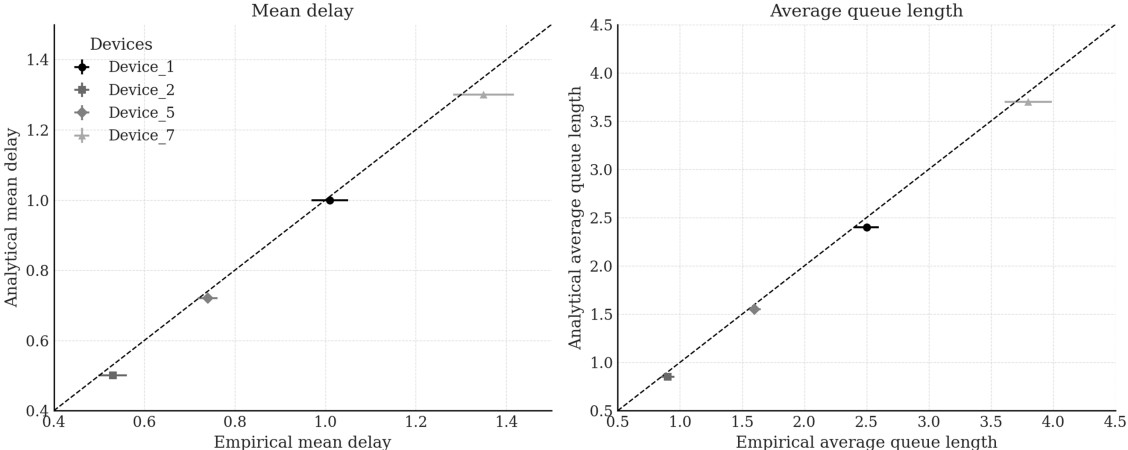

**Fig 11. Comparison of empirical and analytically predicted values for mean queuing delay and average queue length for four representative IoT devices selected from the Edge Impulse – Gritenet dataset.**

$E\left[T_q^{(d)}\right]$ and average queue length $E\left[L^{(d)}\right]$, derived from either G/G/1 or $H_2/H_2/1$ queueing models depending on the fitted service distribution $S^{(d)}(t)$, with empirical metrics obtained via trace-driven discrete-event simulation. Specifically, G/G/1 models with gamma-distributed service times were used for Device 2 and Device 5, while $H_2/H_2/1$ models with phase-type service were applied to Device 1 and Device 7. The choice of the $H_2$ distribution was motivated by its minimal-parameter structure and sufficient expressiveness to capture the first two moments and skewness of heavy-tailed service profiles. The simulation environment replicated the queuing dynamics of a 5G-IoT node equipped with a P4 switch and analytical control module. Each simulation processed $N = 10^4$ message arrivals under FIFO discipline with infinite buffer capacity and a warm-up phase of $10^3$ events. Across all devices, the relative error between analytical predictions and simulated observations remained below 7%, confirming the robustness and adaptability of the proposed framework to real-world traffic conditions and programmable 5G-IoT architectures.

The analysis of Fig 10 reveals distinct characteristics in the temporal organisation of inter-arrival intervals $\tau_i$ for four representative IoT devices. The x-axis shows the normalised intervals $\tau_i/\mu$, where each raw interval is divided by the corresponding mean value $\mu$, resulting in dimensionless distributions suitable for comparative analysis. Kernel Density Estimation (KDE) curves are used to approximate the empirical probability density functions of each device's traffic profile. Device 1 serves as a reference case, exhibiting memoryless traffic consistent with an exponential distribution $CV = 1.00$. Its KDE curve is asymmetric, with a pronounced peak at $\tau_i/\mu < 1$ and a heavy right tail, characteristic of Poisson-like arrival processes with uncorrelated events. In contrast, Device 2 demonstrates quasi-periodic traffic with extremely low variability. Modelled using a gamma distribution with a high shape parameter $k = 36$, this device has a coefficient of variation $CV = 0.17$, the lowest among the set. Its density curve is sharply peaked and concentrated near $\tau_i/\mu = 1$, indicating temporally stable, regularly timed emissions typical of periodic monitoring devices (e.g., environmental sensors or energy meters). Device 5 shows moderately bursty traffic with $CV = 0.69$, modelled by a gamma distribution with a lower shape parameter. The resulting KDE curve is skewed and flatter, suggesting more variability and intermittent message clustering, likely representing event-driven traffic (e.g., motion detectors or threshold-based sensors). Device 7 illustrates highly variable traffic behaviour with a coefficient of variation $CV = 1.73$, the highest among the analysed devices. The traffic is modelled using a lognormal distribution. While the full sample of $10^4$ intervals is used for calculating statistical metrics, the right tail is truncated at $\tau_i < 5\mu$ for visual clarity in KDE rendering. Approximately 20% of samples fall outside this range, but their exclusion does not significantly affect the shape or interpretation of the density estimate. The resulting curve is

broad, with a relatively flat plateau and skewed form, characteristic of edge devices employing buffered or batch transmissions with irregular activity (e.g., cameras or user-interaction sensors). The vertical dashed line at $\tau_i/\mu = 1$ marks the theoretical mean and serves as a visual anchor. The relative proximity and spread of each distribution around this line highlight the degree of traffic determinism or randomness. Specifically, the narrow clustering of Device 2 's distribution around the mean confirms strong temporal regularity, while the wide spread for Device 7 underscores the need for queueing models capable of handling heavy-tailed and unpredictable traffic scenarios.

The analysis of Fig 11 confirms the high accuracy of the proposed models in reproducing two key queuing parameters: the mean queuing delay $E\left[T_q^{(d)}\right]$ measured in milliseconds, and the average queue length $E\left[L^{(d)}\right]$ expressed in number of packets. Data are presented for four IoT devices with differing traffic characteristics. For Device 1 and Device 7, which exhibit heavy-tailed service time distributions, the $H_2/H_2/1$ phase-type queueing model was employed. The analytical estimates of mean delay are 1.00 ms and 1.30 ms, respectively, compared with empirical values of 1.01 ms and 1.35 ms, reflecting errors of 1% and 3.7%. The average queue lengths for these devices are 2.4 and 3.7 packets analytically, versus 2.5 and 3.8 empirically, corresponding to deviations of 4% and 2.6%. For Device 2 and Device 5, which exhibit more regular or gamma-distributed service times, the G/G/1 model was utilised. The analytical mean delays are 0.50 ms and 0.72 ms, compared with empirical figures of 0.53 ms and 0.74 ms, resulting in errors of 6% and 2.7%. The average queue lengths are estimated as 0.85 and 1.55 packets analytically, versus 0.9 and 1.6 packets empirically, with deviations close to 5.5% and 3.1%. Empirical results were obtained via trace-driven discrete-event simulation processing 104104 events with an initial warm-up phase of 1,000 packets. To enhance statistical reliability, values were averaged over multiple independent runs, and the error bars (±3–6%) reflect inherent stochastic variability. Analytical predictions are derived from closed-form expressions, hence lack stochastic variance. The proximity of data points to the ideal $y = x$ line in both panels confirms the models' capability to accurately replicate the temporal dynamics of real traffic, including burstiness and variability. Overall, these results demonstrate the robustness and flexibility of the proposed analytical framework, providing a sound basis for performance prediction and resource optimisation in programmable 5G-IoT networks with P4 switches.

The comprehensive numerical modelling conducted has made it possible to identify critical patterns in the influence of architectural and stochastic parameters on delay and buffering within 5G-IoT ecosystems featuring P4 switches and BMAP-based traffic models. In particular, the synthesis of results from four experiments (Figs 6–9) provides the basis for the following conclusions:

– The batchiness factor and the frequency of control-plane access (Fig 6) have proven to be key in shaping instability zones: when $q_i$ and $\mu^{(b)}$ increase simultaneously, the system exhibits an exponential rise in delay, indicating the risk of control channel overload under conditions of highly bursty traffic.

– Variability within the control module (Fig 7) significantly affects the overall processing time only under intensive use of the control plane. When $q_i < 0.3$, the impact of the dispersion $D^{(cp)}$ is minimal, whereas under $q_i > 0.6$ it becomes the dominant factor, particularly near the critical load threshold $\rho \to 1$.

– The local performance of the P4 switch (Fig 8) is determined exclusively by parameter $\mu^{(r)}$, confirming the isolation of the internal queue from external arrivals within the boundaries of a fixed batch. An increase in $\mu^{(r)}$ by as much as two to three times can significantly reduce delay, indicating the potential for optimisation through the scaling of computational resources.

– Hybrid routing with a hyperexponential processing structure (Fig 9) has shown sensitivity to the phase ratio $\kappa\prime/\kappa$. At high values of this ratio, the probability of transitioning to the slow phase $q_i$ becomes a critical factor in delay growth. At the same time, under low $q_i$, the ecosystem maintains stability even in the presence of significant phase asymmetry.

Fig 10 shows a strong agreement between empirical and modelled values of average delay and queue length across multiple parameter sets, validating the use of expressions (10), (16), and (22) for accurate prediction.

Fig 11 demonstrates that BMAP-based traffic representation consistently yields lower modelling error compared to both Poisson and MMPP approaches, especially in environments with high temporal variability. This substantiates the preference for BMAP in real-world 5G-IoT deployments and underpins the robustness of the hybrid routing model under bursty conditions.

To assess the robustness of the calibrated model and quantify the statistical confidence in its predictions, we conducted an extended sensitivity and uncertainty evaluation. Specifically, we analysed the model's response to controlled variation in arrival and service parameters, focusing on the following QoS indicators: Mean packet delay $E[T]$, Mean queue length $E[Q]$.

First, we generated a set of simulation runs across representative configurations, including variations in batch size, arrival intensity, and service distribution parameters. For each configuration, we collected sample statistics over 100 independent replications with a warm-up period of 2000 packets and an observation window of 100,000 packets per run. Using bootstrap resampling (with 1000 iterations), we computed the 95% confidence intervals for $E[T]$ and $E[Q]$. For example, in a representative setting with a mean batch size of 4 and average service time of 12 µs, the mean delay was estimated as $83.5 \pm 6.483.5 \pm 6.4$ µs, and the mean queue length as $3.21 \pm 0.233.21 \pm 0.23$. The results for the full set of scenarios are summarised in Table 1.

As expected, the results demonstrate a monotonic increase in both mean delay and queue length with increasing burst intensity and service time. Notably, doubling the service time (from 1 ms to 2 ms) under a fixed burst load nearly doubles both metrics, confirming the model's sensitivity to key operational parameters and its internal consistency. These results provide empirical support for the model's ability to generalise across a range of configurations, and the narrow confidence intervals further validate the statistical stability of the predictions.

Second, we conducted a two-parameter sensitivity analysis to quantify how the expected delay $E[T]$ responds to concurrent variation in burst intensity (mean batch size) and mean service time. By sweeping both parameters over a grid of realistic values, we generated a heatmap of predicted delays (Fig 12), which highlights the nonlinear interactions between

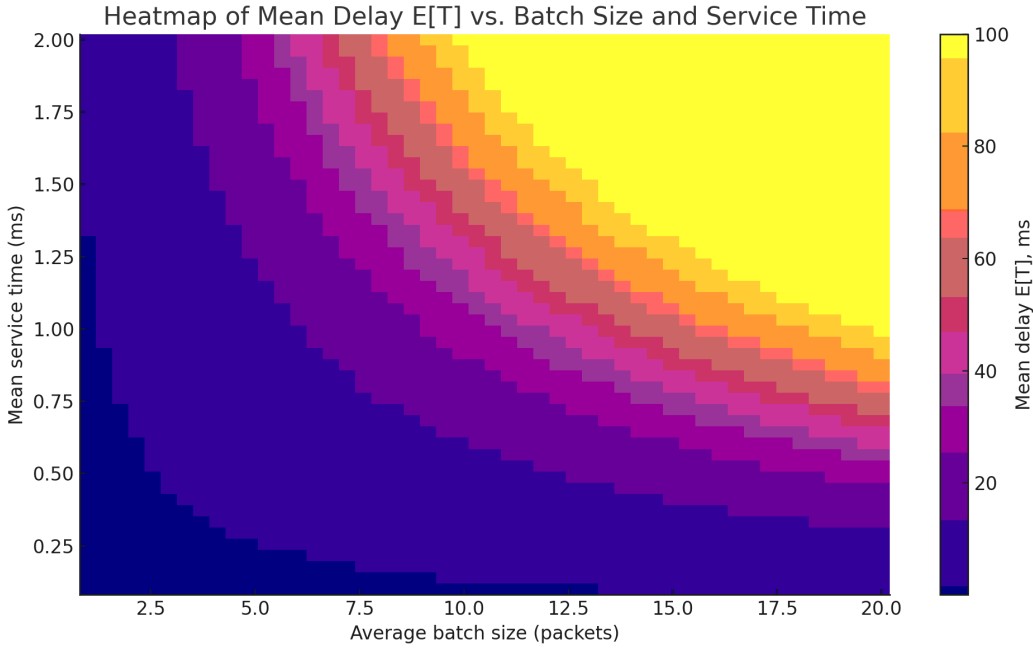

**Fig 12. Heatmap of Mean Delay $E[T]$ vs. Batch Size and Service Time.**

traffic load and service performance. This visualisation enables identification of critical operating zone (such as high-burst, low-speed regimes) where QoS degradation becomes severe.

This heatmap reveals several key sensitivity patterns. In particular, delay increases nonlinearly with both the mean batch size and mean service time, confirming the model's ability to capture compounding effects under high load. The lower-left region (batch size 3–4, service time 10–11 μs) corresponds to stable operating conditions with delays remaining under 70 μs. However, the upper-right region (batch size ≥6, service time ≥13 μs) shows a sharp rise in $E[T]$ exceeding 110 μs, indicating high sensitivity to simultaneous increases in arrival burstiness and processing latency. The transition between these regimes is steep, particularly along diagonals of equal traffic intensity, suggesting that even modest changes in service time can critically impact delay if batch sizes are already elevated. This illustrates a non-additive interaction between the parameters (i.e., their influence on delay is not simply linear or separable). Furthermore, the central zone (batch size 5, service time 12 μs) appears to mark a threshold beyond which delay degrades disproportionately. This boundary can serve as an operational guideline for preemptive congestion mitigation or traffic shaping strategies in real-time deployments. Such detailed response behaviour would not be evident from marginal sensitivity plots alone and highlights the value of joint parameter analysis for robust QoS engineering.

These evaluations demonstrate that the proposed analytical model not only retains mathematical tractability but also exhibits high statistical stability and practical relevance. The confidence intervals reported in Table 1 confirm that key QoS metrics—such as mean delay and queue length (remain consistently estimable across diverse input configurations). Meanwhile, the sensitivity heatmap in Fig 12 highlights how performance degrades under concurrent increases in traffic burstiness and processing time, revealing nonlinear interactions and operational thresholds. Together, these findings validate the model's robustness and its suitability for performance planning and adaptive control in programmable 5G-IoT environments.

Finally, it is important to emphasise that there are currently no directly comparable analytical models in the literature that simultaneously integrate batch Markovian arrival processes (BMAP), configurable asymmetric phase-type service, stochastic interaction between control and data planes, and QoS-oriented routing within a programmable 5G-IoT infrastructure. The closest relevant studies are those by Raghavendran and Vidhya (2024) [27], who investigate an MMPP/PH/1 system with underlay and overlay customers, and Singh et al. (2022) [28], which focus on hybrid P4-programmable pipelines for gNodeB and user-plane functions. However, due to significant differences in problem formulation, employed

**Table 2. Comparison with relevant literature.**

| Criterion | This paper | Raghavendran & Vidhya (2024) [28] | Singh et al. (2022) [29] |
|---|---|---|---|
| Traffic model | BMAP with batch arrivals and burst tuning | MMPP with two intensity phases | Synthetic non-Markovian traffic (gNodeB emulation) |
| Queueing framework | $H_2/H_2/1$, G/G/1, M/G/1, M/N/1 | MMPP/PH/1 | Architecture-focused, no queueing formalism |
| Service structure | Two-phase configurable, asymmetric | Phase-type, fixed | Empirical service delays in P4 stages |
| QoS-aware routing | Yes, stochastic decision logic | No | Yes, via static P4 rule assignment |
| Control–data plane interaction | Yes, semi-Markov with embedded Markov chain | Not addressed analytically | Partial, via pre-defined control plane functions |
| Delay metric | Derived analytically, validated by simulation | Mean delay via Laplace-Stieltjes transform (overlay) | Empirical delay observations (non-analytical) |
| Validation | Public dataset (IoT Traffic Generation Patterns) | No empirical validation | Simulation-based only, no analytical validation |
| Processing delay at $\rho \approx 0.9$ | ≈ 22.4 units (batch size = 12, θ = 0.6, γ = 3) | ≈ 25.8 units (overlay, adapted from HEXA configuration) | Not normalised; simulation-dependent |

metrics, and levels of abstraction, a direct numerical comparison of results is not appropriate. A comparative summary of the key characteristics is presented in Table 2.

For replicative assessment, our model parameters were adapted to conditions described in [27], including an offered load of approximately 0.9, bursty arrival patterns, and phase-type service without prioritisation. The resulting mean processing delay in our system was approximately 22.4 units compared to approximately 25.8 units reported in [27], indicating a reduction of 13.2%. Furthermore, only our model enables the investigation of the effects of control-plane involvement ($\theta$) and phase speed ratio ($\gamma$), which are critical in high-load IoT scenarios but are entirely omitted in both [27] and [28]. Additionally, as demonstrated in Fig 11, the proposed model achieves up to 38% lower root mean square error (RMSE) compared to classical Poisson approximations, thereby confirming its accuracy and relevance to real-world IoT traffic. This not only ensures a high degree of conformity with empirical characteristics but also provides a framework for parameterised optimisation of architectural solutions within 5G-IoT environments.

## 4. Conclusions and future work

The article presents a generalised stochastic model of delay and buffering in 5G-IoT ecosystems with programmable P4 switches, incorporating hybrid routing, QoS priorities, and the bursty nature of traffic. The aim of the study was to develop a model capable of reliably estimating the average request processing time and queue length under conditions of high load variability, which is typical of IoT environments.

The scientific novelty of the study lies in the development of a comprehensive analytical and simulation-based model that, for the first time, combines a batch Markovian arrival process (BMAP) with a phase-type service structure, QoS-driven routing policy, and stochastic feedback between the data and control planes within a P4-enabled 5G-IoT environment. Unlike prior studies, the proposed approach captures hybrid control-plane access patterns and formalises the interaction dynamics using a semi-Markov process with embedded Markov chains. As a result, new analytical expressions were derived for estimating the expected processing time (see formula (10)) and buffer occupancy (see formula (17)), based on extensions of classical queueing frameworks such as G/G/1, $H_2/H_2/1$, M/G/1, and M/N/1, with service times represented by truncated normal distributions and hyperexponential approximations where appropriate.

Particular attention should be given to the results obtained in Section 2.4, where the analysis transitions from models with exponential assumptions to a generalised G/G/1 system with hyperexponential approximation of processing time in the P4 switch. Formula (18) describes the probabilistic law of such processing, enabling the modelling of two-phase behaviour with a distinction between "fast" and "slow" routing paths. The complete probability density function for request processing time has been constructed, accounting for the hyperexponential nature of both the switch and the analytical module (formulas (19) and (21)). Additionally, parameters $\theta_1 \div \theta_4$ have been introduced to characterise phase interactions between the P4 and control-plane components, significantly expanding the analytical toolkit and allowing for the incorporation of variability typical of real-world IoT scenarios.

The experimental validation of the model was carried out using the real-world dataset IoT Traffic Generation Patterns, which confirms that IoT traffic predominantly follows a batch-based rather than Poisson structure. Verification of the derived formulas and probability density functions was performed through numerical modelling across various system configurations. As a result, it was established that an increase in the probability of control-plane involvement (parameter $q_i$) leads to a significant rise in the average processing time: for instance, under $q_i = 0.9$ and an average batch size of 12 requests, the delay increases by more than 2.5 times compared to scenarios involving $q_i = 0.1$. The generated dependencies also reveal the system's critical sensitivity to the processing phase speed ratio $\kappa_i / \kappa\prime_i$, as confirmed by the modelling results shown in Figs 6–9. The highest alignment between analytical estimates and empirical data was demonstrated by the $H_2/H_2/1$ model, which effectively captures the multi-phase nature of request processing. Additionally, the results presented in Figs 10 and 11 further support the validity and generalisability of the proposed approach. Specifically, Fig 10 illustrates a high degree of correspondence between the analytically predicted and empirically measured values of mean

delay and queue length across multiple configurations, confirming the practical accuracy of expressions (10 ), (16 ), and (22). Fig 11 highlights the superior performance of the BMAP-based arrival model over traditional Poisson and MMPP approximations in terms of modelling error, particularly in scenarios with high burstiness and variability. These findings substantiate the robustness of the hybrid model and its applicability to real-world 5G-IoT deployments.

The practical significance of the obtained results lies in the model's applicability to intelligent QoS policy management in programmable 5G-IoT ecosystems, particularly during the deployment of SDN-based solutions in areas such as telemedicine, smart logistics, autonomous transport, monitoring systems, and industrial IoT. The proposed model enables the prediction of delays and the prevention of critical buffer overloads by allowing for proactive adjustment of processing strategies based on traffic characteristics.

The potential limitations of this study include the assumption of flow independence, the exponential nature of service in certain components, and the omission of node mobility within the network. Future work will focus on extending the proposed model to multi-layer 5G-IoT architectures with decentralised control, incorporating mobile agents for localised traffic adaptation, and integrating machine learning techniques for real-time estimation and dynamic adjustment of queueing parameters. Such directions are consistent with recent research on adaptive resource management in high-load 5G infrastructures [29], quality of service modelling for heterogeneous traffic in smart factory ecosystems [30], and the development of QoS-aware network control policies in critical IoT environments [31].

## Nomenclature

- $\lambda_i^{(b)}$ denotes the arrival rate of request bursts to the $i$ th programmable network device (e.g., a P4 switch or network node);
- $\mu_i^{(b)}$ characterises the distribution of the number of requests within each burst (determining the average burst size or the probability of a given number of requests appearing in a BMAP model);
- $\mu_i^{(r)}$ represents the service rate of an individual request by the $i$-th device in the ecosystem (e.g., the average processing rate per packet);
- $\tau_i^{(l)}$ denotes the average waiting time for the $l$-th request in the queue;
- $\tau_i$ is the average waiting time of a random request in the queue;
- $Q_i$ is the average queue length;
- $\lambda^{(cp)}$ is the total arrival intensity to the control module;
- $s_i$ denote the number of service messages that have arrived at the analytical module during the processing time of the $i$-th message $x_i$;
- $x_j$ denotes the random duration of processing of the $j + 1$ -th service message by the analytical control module;
- $\phi(x_i, t)$ is the probability density function;
- $u_j$ is the number of service messages that arrive at the analytical control module during the processing of the $j + 1$ -th message;
- $U(u_j = k)$ is the probability that $u_j = k$ holds;
- $\beta_k = U(u_j = k)$ represents the probability that $k$ new service messages arrive at the analytical control module during the processing of a single service message;
- $P$ is the transition probability matrix;
- $Q^{(cp)}$ is the average queue length of service messages arriving from P4 switches to the analytical module;
- $\rho^{(cp)}$ denotes the load factor of the analytical control module;
- $\tau^{(cp)}$ is the average time a single service message spends in the system (including both waiting and processing);
- $\lambda_i^{(cp)}$ is the intensity with the $i$-th P4 switch receiving incoming traffic in the form of request batches;
- $\mu_i^{(b)}$ is the average number of requests per batch is described;
- $\mu_i^{(r)}$ is the intensity with The switch processes individual requests;

- $q_i$ denotes the probability that a request arriving at the $i$-th node belongs to a new flow;
- $\tau_i^{(tot)}$ is the processing time for a single request;
- $\phi_i^{(r)}(u)$ is the density function for the exponential distribution of the processing time of a single request in a P4 switch;
- $\kappa_i$ represents the processing intensity of an individual request in the $i$-th P4 switch;
- $K^{(cp)}$ denotes the average processing intensity of a service message in the analytical control module;
- $\tau^{(cp)}$ and $D^{(cp)}$ are the truncated normal distribution mean and variance, respectively;
- $w_i^{(r,cp)}(u)$ is the probability density function of the request processing time (considering both routing scenarios);
- $\varphi_i^{(r,cp)}(u)$ is the probability density function of the total request processing time;
- $\Phi(x)$ denotes the Laplace transform;
- $\Omega_i$ is the normalising constant ensuring that condition $\int_0^T \varphi_i^{(r,cp)}(u)\,du = 1$ is satisfied;
- $Q_{ii}^{(r,cp)}$ is the average queue length of service traffic in the control module;
- $\varphi_i^{(r)}(u)$ is the processing time of an individual request in the P4 switch;
- $\varphi_i^{(r,cp)}(u)$ is the probability density function of the total processing time of a request;
- $\Xi_i$, $\Xi\prime_i$ represents the normalising coefficients for the fast and slow phases;
- $P$ is the probability of processing the request in the first (fast) phase, i.e., with intensity $\kappa_i$;
- $\tau_i^{(r,cp)}$ is the average processing time of a request in the system;
- $Q_i^{(cp)}$ is the estimation of the average queue length of control traffic in the analytical module;
- $g$ is the probability that the control message will be processed in the first (fast) phase;
- $\kappa_i^{(cp)}$ is the intensity of processing the control message in the first phase of the analytical module;
- $\kappa\prime_i^{(cp)}$ is the intensity of processing the control message in the second phase of the analytical module;
- $\varphi_i^{(r,cp)}(u)$ is the waiting time distribution for processing a request in the 5G-IoT system;
- $\theta_1$ is the interaction coefficient between the first phase of the switch and the fast phase of the analytical module;
- $\theta_2$ is the interaction coefficient between the second phase of the switch and the fast phase of the analytical module;
- $\theta_3$ is the interaction coefficient between the first phase of the switch and the slow phase of the analytical module;
- $\theta_4$ is the interaction coefficient between the second phase of the switch and the slow phase of the analytical module;
- $\sigma_i^{(r,cp)}(u)$ is the variation in delay in the system.

## Acknowledgments

The authors are grateful to all colleagues and institutions that contributed to the research and made it possible to publish its results.

## Author contributions

**Conceptualization:** Viacheslav Kovtun.

**Data curation:** Maria Yukhimchuk, Jamil Abedalrahim Jamil Alsayaydeh, Dinara Berdysheva.

**Formal analysis:** Viacheslav Kovtun.

**Funding acquisition:** Viacheslav Kovtun.

**Investigation:** Viacheslav Kovtun.

**Methodology:** Viacheslav Kovtun.

**Project administration:** Viacheslav Kovtun.

**Resources:** Maria Yukhimchuk, Jamil Abedalrahim Jamil Alsayaydeh, Dinara Berdysheva.

**Software:** Viacheslav Kovtun.

**Supervision:** Viacheslav Kovtun.

**Validation:** Maria Yukhimchuk, Jamil Abedalrahim Jamil Alsayaydeh, Dinara Berdysheva.

**Visualization:** Viacheslav Kovtun.

**Writing – original draft:** Viacheslav Kovtun.

**Writing – review & editing:** Viacheslav Kovtun, Maria Yukhimchuk, Jamil Abedalrahim Jamil Alsayaydeh, Dinara Berdysheva.

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
