## [Decision Letter · Decision Letter 0]

8 Jul 2025

PONE-D-25-30011Stochastic modelling of delays and buffering in 5G-IoT ecosystems with programmable P4 switches based on BMAPPLOS ONE

Dear Dr. Kovtun,

Thank you for submitting your manuscript to PLOS ONE. After careful consideration, we feel that it has merit but does not fully meet PLOS ONE’s publication criteria as it currently stands. Therefore, we invite you to submit a revised version of the manuscript that addresses the points raised during the review process.

Please, address the concerns raised by the reviewers in a revised manuscript. Both reviewers identified issues regarding some assumptions made during modelling phase (e.g., the assumption of infinite buffers) and some shortcomings of the experiments (e.g., the utilization of a truncated normal distribution). In the revised version, please included other experiments beyond the ones utilizing the Kaggle dataset. As observed by the reviewers, the authors are incentivized to perform experiments using real testbed or emulated ones. 

We look forward to receiving your revised manuscript.

Kind regards,

Ivan Zyrianoff

Academic Editor

PLOS ONE

Journal Requirements: 

3. In the online submission form, you indicated that your data is available only on request from a third party. Please note that your Data Availability Statement is currently missing contact details for the third party, such as an email address or a link to where data requests can be made. Please update your statement with the missing information.

Reviewers' comments:

Reviewer's Responses to Questions

**Comments to the Author**

1. Is the manuscript technically sound, and do the data support the conclusions?

Reviewer #1: Yes

Reviewer #2: Partly

2. Has the statistical analysis been performed appropriately and rigorously? 

Reviewer #1: Yes

Reviewer #2: Yes

3. Have the authors made all data underlying the findings in their manuscript fully available?

Reviewer #1: Yes

Reviewer #2: Yes

4. Is the manuscript presented in an intelligible fashion and written in standard English?

Reviewer #1: Yes

Reviewer #2: Yes

5. Review Comments to the Author

Reviewer #1: The manuscript presents an original and timely approach by combining BMAP, phase-type service models, and semi-Markov processes to model delays and buffering in 5G-IoT networks with programmable P4 switches. This combination is innovative and offers a more realistic view of IoT traffic behavior, especially under bursty conditions that classical models often fail to capture.

Another strong point is the analytical rigor and empirical validation. The authors derive meaningful expressions for the average delay and queue length, and validate them using real-world data. The comparison with Poisson and MMPP-based models clearly demonstrates the improved accuracy of the proposed approach, which has practical implications for QoS in IoT networks.

One major issue, in this reviewer opinion, lies in the use of 3D visualizations to present results. While intended to show parameter interactions, these plots often hinder clarity. In particular, Figs 6 and 8 can be replaced without loss, and Figs 7 and 9 don't bring any clarity to the results. Replacing them with 2D plots (e.g., heatmaps or line graphs) would improve the reader's ability to interpret trends and compare outcomes.

Additionally, assumptions such as infinite buffers and simplified traffic sources may not accurately reflect real-world constraints. A broader evaluation with diverse traffic scenarios and comparison to other modeling techniques would help confirm the model's general applicability.

Some open questions:

How were the main model parameters (like BMAP rates and service times) estimated from the real dataset? Did you use any fitting method?

What is the impact of assuming infinite buffers? Would the results still hold under more realistic buffer limits?

Why did you choose a truncated normal distribution for control-plane processing time? Was this based on real data, or was it used for simplicity?

Can your model handle other types of IoT traffic beyond the Kaggle dataset? Have you tested it on different scenarios?

Reviewer #2: Thank you for the opportunity to review this manuscript. The paper addresses an important and technically complex challenge in modern network design: accurately modeling delay and buffering behavior in 5G-IoT environments using programmable P4 switches. The authors propose a hybrid stochastic framework that combines BMAP-based traffic arrivals, phase-type service distributions, and semi-Markov modeling of control-plane interactions. This integration is original and mathematically rigorous, and the theoretical formulation demonstrates a deep understanding of queueing theory and its application to programmable networks.

That said, there are several areas that require substantial revision before the manuscript can be considered for publication.

One key concern is the limited practical validation. While the use of real-world traffic data from the Kaggle repository is appreciated, the results are derived entirely from simulation and analytical modeling. There is no testbed evaluation, no experimental setup with actual P4 environments (even emulated), and no prototype or application scenario that demonstrates how the model would be used in practice. For a topic so closely tied to performance engineering, this disconnect between theory and application significantly weakens the impact of the results.

Additionally, several assumptions in the model are overly idealized. For instance, the use of infinite buffer capacities and the assumption of no packet loss may simplify the derivations, but they also distance the analysis from real-world constraints in IoT networks. While the authors argue that such assumptions ensure analytical tractability, a discussion of their practical limitations—or a sensitivity analysis to approximate more constrained environments—would be highly beneficial.

From a statistical perspective, the modeling choices are mostly appropriate, but the justification for specific distributions, such as the truncated normal used to model control-plane processing delays, is insufficient. It is not clear whether this choice was based on data fitting, prior literature, or simply mathematical convenience. Moreover, the paper would benefit from including confidence intervals or a brief discussion of model sensitivity to parameter variations.

Regarding presentation, the manuscript is generally well-written in standard English, but the clarity of the exposition could be improved. Some sections—particularly the mathematical modeling—are quite dense and could be difficult to follow for non-specialists. Including a table of notation and a summary of key variables would help significantly. Additionally, restructuring some of the longer paragraphs and simplifying the transitions between mathematical and conceptual descriptions would improve the overall readability.

In summary, this work offers a promising and original contribution to the field, and the modeling approach has merit. However, in its current form, the paper lacks sufficient empirical grounding and clarity to meet the publication standards of PLOS ONE. I encourage the authors to revise the manuscript thoroughly, addressing the concerns outlined above. With additional work to strengthen the validation, clarify assumptions, and improve presentation, this manuscript could become a valuable resource for researchers and engineers working with P4-based 5G-IoT systems.

6. PLOS authors have the option to publish the peer review history of their article (what does this mean? ). If published, this will include your full peer review and any attached files.

**Do you want your identity to be public for this peer review?** For information about this choice, including consent withdrawal, please see our Privacy Policy .

Reviewer #1: No

Reviewer #2: No

---

## [Author Response · Author response to Decision Letter 1]

10 Jul 2025

Our responses to the reviewers' comments are provided in a separate file as part of the submission.

---

## [Decision Letter · Decision Letter 1]

4 Aug 2025

Stochastic modelling of delays and buffering in 5G-IoT ecosystems with programmable P4 switches based on BMAP

PONE-D-25-30011R1

Dear Dr. Kovtun,

We’re pleased to inform you that your manuscript has been judged scientifically suitable for publication and will be formally accepted for publication once it meets all outstanding technical requirements.

Kind regards,

Ivan Zyrianoff

Academic Editor

PLOS ONE

Additional Editor Comments (optional):

please, address the reviewers comments in the camera-ready version. The comments made are small modifications on to the manuscript. 

Reviewers' comments:

Reviewer's Responses to Questions

**Comments to the Author**

1. If the authors have adequately addressed your comments raised in a previous round of review and you feel that this manuscript is now acceptable for publication, you may indicate that here to bypass the “Comments to the Author” section, enter your conflict of interest statement in the “Confidential to Editor” section, and submit your "Accept" recommendation.

Reviewer #1: All comments have been addressed

Reviewer #2: All comments have been addressed

2. Is the manuscript technically sound, and do the data support the conclusions?

Reviewer #1: Yes

Reviewer #2: Yes

3. Has the statistical analysis been performed appropriately and rigorously? 

Reviewer #1: Yes

Reviewer #2: Yes

4. Have the authors made all data underlying the findings in their manuscript fully available?

Reviewer #1: Yes

Reviewer #2: Yes

5. Is the manuscript presented in an intelligible fashion and written in standard English?

Reviewer #1: Yes

Reviewer #2: Yes

6. Review Comments to the Author

Reviewer #1: I would like to thank you in advance for the modifications made to the paper, as well as the explanations provided in your response letter. I believe the paper has reached a level of maturity suitable for publication.

Just as a suggestion for the possible final version of the article, I recommend a visual evaluation of Table 2, modifying the font size or including horizontal lines for easier viewing, following the journal's guidelines, of course.

Optionally, the list of nomenclatures could contain more concise explanations respecting the width of the page, thus reducing its length - particularly a table would be nice.

Reviewer #2: The authors have addressed the concerns raised in the previous review round with care and precision. The manuscript is now technically sound, clearly presented, and consistent in its arguments.

The most significant limitation pointed out earlier — the lack of empirical validation — was directly addressed. Although physical experiments have not yet been executed, the revised version presents a well-structured and feasible roadmap for both emulated and hardware-based validation, including P4-based testbeds and hardware (Intel Tofino 2 and NetFPGA). This plan enhances the credibility of the work and shows commitment to extending the theoretical findings with practical confirmation in the near future.

Regarding the modeling assumptions, especially the infinite buffer scenario and the use of truncated normal distributions for the control plane latency, the authors provided adequate justification and discussed how those choices reflect trade-offs between tractability and realism. Additionally, they acknowledged the limitations and indicated how they plan to refine these assumptions in future work. The statistical modeling, parameter estimation, and explanation of autocorrelation fitting have all been expanded and clarified.

The manuscript also benefits from improved clarity: sections with heavy mathematical content are now easier to follow, the notation has been consolidated, and the flow of the argument is more coherent. The response demonstrates thoughtful engagement with the feedback and an effort to raise the quality of both the theoretical model and its exposition.

7. PLOS authors have the option to publish the peer review history of their article (what does this mean? ). If published, this will include your full peer review and any attached files.

**Do you want your identity to be public for this peer review?** For information about this choice, including consent withdrawal, please see our Privacy Policy .

Reviewer #1: No

Reviewer #2: No

---

## [Editor Report · Acceptance letter]

PONE-D-25-30011R1

PLOS ONE

Dear Dr. Kovtun,

I'm pleased to inform you that your manuscript has been deemed suitable for publication in PLOS ONE. Congratulations! Your manuscript is now being handed over to our production team.

Kind regards,

on behalf of

Mr. Ivan Zyrianoff

Academic Editor

PLOS ONE